# Scaling Laws for Gradient Descent and Sign Descent for Linear Bigram Models under Zipf's Law

**Frederik Kunstner**
frederik.kunstner@inria.fr

**Francis Bach**
francis.bach@inria.fr

## Abstract

Recent works have highlighted optimization difficulties faced by gradient descent in training the first and last layers of transformer-based language models, which are overcome by optimizers such as Adam. These works suggest that the difficulty is linked to the heavy-tailed distribution of words in text data, where the frequency of the $k$th most frequent word $\pi_k$ is proportional to $1/k$, following Zipf's law. To better understand the impact of the data distribution on training performance, we study a linear bigram model for next-token prediction when the tokens follow a power law $\pi_k \propto 1/k^\alpha$ parameterized by the exponent $\alpha > 0$. We derive optimization scaling laws for deterministic gradient descent and sign descent as a proxy for Adam as a function of the exponent $\alpha$. Existing theoretical investigations in scaling laws assume that the eigenvalues of the data decay as a power law with exponent $\alpha > 1$. This assumption effectively makes the problem "finite dimensional" as most of the loss comes from a few of the largest eigencomponents. In comparison, we show that the problem is more difficult when the data have heavier tails. The case $\alpha = 1$ as found in language is "worst-case" for gradient descent, in that the number of iterations required to reach a small relative error scales almost linearly with dimension. While the performance of sign descent also depends on the dimension, for Zipf-distributed data the number of iterations scales only with the square-root of the dimension, leading to a large improvement for large vocabularies.

## 1 Introduction

Recent works have shown that one of the primary benefits of Adam (Kingma and Ba, 2015) in training transformed-based language models (Vaswani et al., 2017) lies in how it handles the first and last layers (Zhang et al., 2025; Zhao et al., 2025). For language models, the input and output dimensions correspond to distinct words in the vocabulary, where the $k$th most frequent word has frequency $\pi_k \propto 1/k$ following Zipf's law (Piantadosi, 2014). Kunstner et al. (2024) provide evidence that this heavy-tailed distribution leads to optimization difficulties for gradient descent that Adam is able to overcome. They argue that Zipf's law is "worst-case" in that it combines a large imbalance in frequencies, while decaying slowly enough that most samples come from the tail.

Our objective is to formalize this empirical observation, and to describe the impact of the heavy-tailedness of the data distribution on the convergence of gradient descent (GD) and sign descent (SD) as a proxy for Adam (Tieleman and Hinton, 2012; Bernstein et al., 2018; Balles et al., 2020; Chen et al., 2023). We consider a linear bigram model for next-token prediction trained with the square loss, where the token frequencies $\pi_k$ follow a power law $\pi_k \propto 1/k^\alpha$ with exponent $\alpha > 0$. While this problem could be solved directly rather than with iterative methods, it is a good starting point for the theoretical investigation of optimization dynamics. Despite its apparent simplicity, this model already reproduces the observation that GD performs poorly on Zipf-distributed data (see Fig. 1). The behavior of gradient and sign descent are also not well described by current results, see Section 1.2.

Our approach is inspired by the line of work on theoretical scaling laws, also known as asymptotic convergence as the dimensionality grows (e.g., Caponnetto and De Vito, 2007; Advani et al., 2020;

39th Conference on Neural Information Processing Systems (NeurIPS 2025).

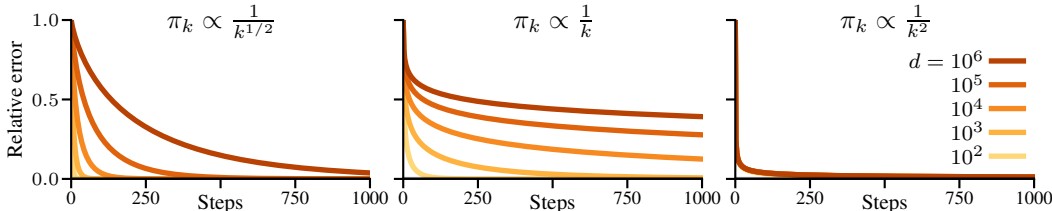

Figure 1: **Gradient descent (GD) scales badly with vocabulary size when the data is Zipfian.**
Relative error on a linear bigram problem with squared loss trained with GD with vocabulary size $d$
when the word frequencies follow $\pi_k \propto 1/k^\alpha$. For $\alpha \leq 1$ (left, middle) the performance degrades
with vocabulary size and is worst for Zipf-distributed data ($\alpha = 1$). When the frequencies have lighter
tails ($\alpha = 2$, right) GD works well for all vocabulary sizes. Our objective is to explain this behavior.

Berthier et al., 2020; Bahri et al., 2021; Cui et al., 2021; Maloney et al., 2022; Paquette et al., 2024).
Instead of analyzing the generalization error of online gradient descent as the dimension of the model
and sample size grow, we study the convergence rate of GD as the dimension grows. Spectral assump-
tions on the eigenvalues of the Hessian following a power-law are common in the literature, which in
our case correspond to assumptions on word frequencies. But these works focus on power-laws that
are not "too" heavy-tailed, $1/k^\alpha$ with $\alpha > 1$, which lead to sublinear rates independent of dimension.
In contrast, we focus on the case $\alpha \leq 1$ where it becomes impossible to make progress unless the
number of iterations grows with the dimension of the problem. Our contributions are as follows.

1. We propose a simplified model of the word frequencies that leads to tractable dynamics for GD
   and SD, that captures the difference in their training performance experimentally (Fig. 2).

2. We derive scaling laws for GD and SD in this model as a function of $\alpha > 0$, covering power-laws
   that decrease as slow or slower than Zipf's law ($\alpha \leq 1$). This setting is often ignored in existing
   analyses and leads to qualitatively different results, with scaling laws that are not power-laws
   and require the number of iterations $t$ to grow with $d$.

3. For GD on Zipf-distributed data ($\alpha = 1$) the number of iterations required to reach a small
   relative error scales almost linearly with dimension, $t \sim d$. This setting is "worst-case" in that the
   case $\alpha < 1$ results in a better scaling of $t \sim d^\alpha$, while $\alpha > 1$ does not require $t$ to scale with $d$.

4. In comparison, SD under Zipf-distributed data only requires $t \sim \sqrt{d}$, provably confirming its
   benefits over GD on a language task, but not in all settings as SD exhibits worse scaling if $\alpha > 1$.

## 1.1 Overview of the results

We consider a simplified language modeling tasks, given a vocabulary of $d$ words, we train a linear
bigram model with square loss to predict the next word $y \in [d]$ given the current word $x \in [d]$,
represented as one-hot vectors $\mathbf{x}, \mathbf{y} \in \{0, 1\}^d$. The dynamics of the problem depends on the word
frequencies $\pi_k$ and conditional frequencies $\pi_{k \mid j}$, which we assume follow a power law $1/k^\alpha$,
formalized in Section 2. The scaling we analyze is how the loss changes as the dimension $d$ and
number of iterations $t$ increases, depending on $\alpha$. For GD on $\alpha > 1$, we recover the result that the
loss after $t$ steps, $\mathcal{L}_d(t)$, follows the power law $\mathcal{L}_d(t) \sim c_1 t^{-p} + c_2$ as $d \to \infty$ for some power $p$ and
constants $c_1, c_2$. Equivalently, we could write the relative optimality gap as

$$r_d(t) := \frac{\mathcal{L}_d(t) - \mathcal{L}_d^*}{\mathcal{L}_d(0) - \mathcal{L}_d^*} \sim t^{-p},$$

where $\sim$ denotes asymptotic equivalence, $r_d(t) \sim t^{-p}$ means $\lim_{d \to \infty} r_d(t)/t^{-p} = 1$. This rate is
independent of $d$, but specific to GD with $\alpha > 1$. Our results show that in other settings, the number
of iteration $t$ needs to scale with $d$ to achieve an $\varepsilon$-relative optimality gap, $r_d(t) \sim \varepsilon$. The following
is a simplification of our main results, made formal in Theorems 3.1 and 4.5.

**Informal theorem.** *To reach an $\varepsilon$-relative optimality gap, the number of iterations $t$ should scale as*

$$t \asymp \begin{cases} d^\alpha & \text{if } \alpha < 1, \\ d^{1-\epsilon} & \text{if } \alpha = 1, \quad \text{for GD,} \\ 1 & \text{if } \alpha > 1, \end{cases} \qquad \text{and} \qquad t \asymp \begin{cases} 1 & \text{if } \alpha < 1/2, \\ \sqrt{d^{1-\varepsilon}} & \text{if } \alpha = 1/2, \quad \text{for SD,} \\ \sqrt{d} & \text{if } \alpha > 1/2, \end{cases}$$

*where $t \asymp f$ (equivalently, $t = \Theta(f)$) hides constants and factors that depend on $\epsilon$ but not on $d$.*

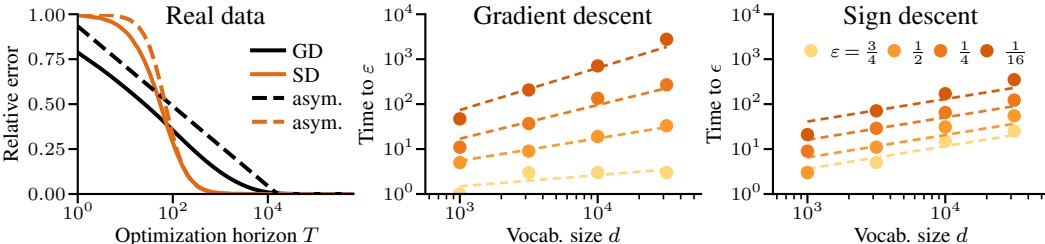

Figure 2: **Our scaling predicts the behavior of gradient descent and sign descent on real data.** Left: the convergence of gradient descent (GD) and sign descent (SD) is close to our asymptotic prediction ( $\boldsymbol{--}$ , $\boldsymbol{--}$ ) on a bigram model with 32k tokens on OpenWebText, although not exactly due to the finite dimension and our simplified model of the frequencies in Assumption 2.3. Middle/Right: as $d$ grows, the number of iterations required to reach $\varepsilon$ relative error matches our predictions, showing that SD scales better with dimension for small $\varepsilon$. We show results on real data (dots) against the scaling of $c_1 d^{1-\varepsilon}$ for GD and $c_2 d^{1/2}$ for SD (dashes) where $c_1, c_2$ are fit to the data.

For language and Zipf-distributed data ($\alpha = 1$), our scaling predicts that the number of iterations required to reach $\varepsilon$ relative error scales almost linearly with $d$ for GD if $\varepsilon$ is small while it scales as $d^{1/2}$ for SD. For common vocabulary sizes of $d = 10^4$ tokens, this leads to a 100-times speedup. Zipf-distributed data is also "worst-case" for GD, as other values of $\alpha$ lead to better scaling in $d$.

We recover the power-law scaling for GD when $\alpha > 1$, but obtain different functional forms for other settings (Theorems 3.1 and 4.5). For $\alpha = 1$, the relative optimality gap behaves as

$$r_d(t) \approx 1 - \frac{\log(2t)}{\log(d)} \quad \text{in the sense that} \quad r_d(\tfrac{1}{2}d^{1-\epsilon}) \sim \epsilon.$$

We confirm these predictions experimentally using real data on OpenWebText, shown in Fig. 2.

## 1.2 Related work

**Convergence of Adam and sign descent.** The benefit of Adam has been argued to stem from its similarity to sign descent, in that the updates are uniform across coordinates (Bernstein et al., 2018; Balles et al., 2020; Chen et al., 2023). This "scale-freeness" can reduce the dependence on the condition number (Zhuang et al., 2022), but this does not imply SD outperforms GD as known convergence rates for sign-like methods instead depend on the dimension $d$ (e.g., Safaryan and Richtárik, 2021; Das et al., 2024; Liu et al., 2025). In the bigram problem with Zipf-distributed data, the dimension grows faster than the condition number, leading to worse guarantees for SD. We compare our asymptotic analysis to existing rates in Appendix B.

**SDE approximations of sign methods.** Scaling laws have been derived for online sign-like algorithms through stochastic differential equations (Ma et al., 2021; Malladi et al., 2022; Xiao et al., 2024; Compagnoni et al., 2025). The focus of these works is on the scaling of the step-size with batch size and the asymptotic stationary distribution of the algorithm which controls the generalization error. As noise is not necessary to reproduce the performance gap between GD and Adam (Kunstner et al., 2023), we instead focus on the impact of heavy-tailed data on the deterministic dynamics.

**Scaling laws and asymptotic results.** Empirical scaling laws have been developed to extrapolate the performance of deep networks at scale and how to balance compute across model and data sizes (Rosenfeld et al., 2020; Kaplan et al., 2020; Hoffmann et al., 2022). Many works have contributed to the theoretical understanding of this scaling behavior through high dimensional analyses and random matrix theory (Advani et al., 2020; Bahri et al., 2021; Maloney et al., 2022; Bordelon et al., 2024a; Lin et al., 2024; Paquette et al., 2024), classical source/capacity conditions from learning theory (Caponnetto and De Vito, 2007; Berthier et al., 2020; Cui et al., 2021), also used in an optimization context (Velikanov and Yarotsky, 2024). However, those works study problems where the spectrum decays fast, and does not cover case $\alpha \leq 1$. This regime, covering Zipf's law, might be more relevant when considering scaling the vocabulary size, as in the work of Gowda and May (2020) and Tao et al. (2024). While they hypothesize that larger vocabularies might lead to worse performance due to overfitting, as larger vocabularies implies fewer examples per word in addition to more compute per step, we show that larger vocabulary size might also need more steps to get the training error down. Closest to our work is perhaps the blog post of Bulatov (2023), which argues that

the loss under GD should approximately behave as $-\log(t/d)$ on a problem matching our setting with $\alpha = 1$. Our work provides a formal justification for this scaling.

## 2 Problem setup

In this section, we present the problem setting, a linear bigram model with square loss, the modeling assumptions used to make the problem tractable and the approach we use to derive our results.

**Problem 2.1** (Linear bigram model). *Let $\mathbf{x}_i, \mathbf{y}_i \in \{0,1\}^d$ for $i = 1, \ldots, n$ be one-hot encodings from $d$ classes (or tokens), with their concatenation $\mathbf{X}, \mathbf{Y} \in \{0,1\}^{n \times d}$, fit with a linear model,*

$$\mathcal{L}_d(\mathbf{W}) = \frac{1}{2n}\|\mathbf{XW} - \mathbf{Y}\|_F^2 \quad \text{where} \quad \mathbf{W} \in \mathbb{R}^{d \times d}, \quad \text{and} \quad \|\mathbf{X}\|_F^2 = \text{Tr}(\mathbf{X}^\top \mathbf{X}).$$

*We define $\pi_k$ and $\pi_{k\,|\,j}$ as the frequencies and conditional frequency statistics of the data,*

$$\pi_k := \frac{1}{n}\sum_{i=1}^{n}\mathbb{1}_{x_i=k}, \quad \pi_{k\,|\,j} := \frac{\sum_{i=1}^{n}\mathbb{1}_{y_i=k}\mathbb{1}_{x_i=j}}{\sum_{i=1}^{n}\mathbb{1}_{x_i=j}}, \quad (\text{with the convention } 0/0 = 0) \quad \forall j, k \in [d].$$

The analysis of GD on quadratics typically uses an eigenvalue decomposition. Consider a $d$-dimensional quadratic $f(\mathbf{x}) = \frac{1}{2}(\mathbf{x}-\mathbf{x}^*)^\top \mathbf{A}(\mathbf{x}-\mathbf{x}^*)$ with minimizer $\mathbf{x}^*$ where the eigenvalues/vectors pairs of $\mathbf{A}$ are $(\lambda_i, \mathbf{v}_i) \in \mathbb{R} \times \mathbb{R}^d$ for $i = 1, \ldots, d$. The dynamics of GD with step-size $\eta$, $\mathbf{x}_{t+1} = \mathbf{x}_t - \eta\mathbf{A}(\mathbf{x}_t - \mathbf{x}^*)$, decompose in terms of the distance along eigenvectors, $\delta_i(t) = \langle \mathbf{v}_i, \mathbf{x}_t - \mathbf{x}^* \rangle$, as

$$f(\mathbf{x}_t) = \frac{1}{2}(\mathbf{x}_0 - \mathbf{x}^*)^\top(\mathbf{I} - \eta\mathbf{A})^t\mathbf{A}(\mathbf{I} - \eta\mathbf{A})^t(\mathbf{x}_0 - \mathbf{x}^*) = \frac{1}{2}\sum_{i=1}^{d}\lambda_i(1 - \eta\lambda_i)^{2t}\delta_i(0)^2.$$

For the $d^2$-dimensional bigram model (2.1), the dynamics depend on the frequencies $\pi_k$ and $\pi_{k\,|\,j}$.

**Proposition 2.2.** *The dynamics of gradient descent on Problem 2.1 initialized at $\mathbf{W} = 0$ are described by the eigenvalues and distances to solution $\lambda_{ij}, \delta_{ij}(0)$ for $i, j = 1, \ldots, d$, using $\mathcal{L}_d(t) = \mathcal{L}_d(\mathbf{W}_t)$,*

$$\mathcal{L}_d(t) - \mathcal{L}_d^* = \frac{1}{2}\sum_{i=1}^{d}\sum_{j=1}^{d}\lambda_{ij}(1 - \eta\lambda_{ij})^{2t}\delta_{ij}(0)^2 = \frac{1}{2}\sum_{i=1}^{d}\pi_i(1 - \eta\pi_i)^{2t}\sum_{j=1}^{d}\pi_{j\,|\,i}^2, \quad (1)$$

*where $\mathcal{L}_d^* = \min \mathcal{L}_d$, as the eigenvalues and distances to solution are $\lambda_{ij} = \pi_i$ and $\delta_{ij}(0)^2 = \pi_{j\,|\,i}^2$.*

*Proof sketch.* The Hessian of $\mathcal{L}_d$ is diagonal, with diagonal blocks $\mathbf{X}^\top\mathbf{X}/n = \text{Diag}([\pi_1, \ldots, \pi_d])$ repeated $d$ times. The eigenvectors are the standard basis with eigenvalues $\lambda_{ij} = \pi_i$. The solution is at $\mathbf{W}^* = (\mathbf{X}^\top\mathbf{X})^{-1}\mathbf{X}^\top\mathbf{Y}$, where $w_{ij}^* = \pi_{j\,|\,i}$ as $[\mathbf{X}^\top\mathbf{Y}/n]_{ij} = \pi_{j\,|\,i}\pi_i$, giving $\delta_{ij}(0)^2 = \pi_{j\,|\,i}^2$. $\quad\square$

### 2.1 Modeling assumptions

Getting an interpretable form of the rate in Eq. (1) requires assumptions on the values of $\lambda_i$ and $\delta_i$. Assuming $\mu \leq \lambda_i \leq L$ leads to the typical smooth (strongly-)convex rates (e.g., Nesterov, 2018),

$$\mathcal{L}_d(t) - \mathcal{L}_d^* \leq \frac{L}{t}\sum_{i=1}^{d}\delta_i(0)^2, \qquad \mathcal{L}_d(t) - \mathcal{L}_d^* \leq \left(1 - \frac{\mu}{L}\right)^t(\mathcal{L}_d(0) - \mathcal{L}_d^*).$$

While valid, these worst-case bounds are too coarse to capture the richness of the behavior of GD and becomes vacuous if $\mu \to 0$ or $\sum_{i=1}^{d}\delta_i(\mathbf{w}_0)^2 \to \infty$ as $d \to \infty$. To obtain fine-grained results, we assume that the frequencies $\pi_k$ and conditional frequencies $\pi_{k\,|\,j}$ follow power laws.

**Assumption 2.3** (Heavy-tailed data). *We assume that the frequencies and conditional frequencies follow a frequency-rank power law with exponent $\alpha > 0$. That is, assuming the frequencies are sorted ($\pi_k \geq \pi_{k+1}$) and defining the sorting permutations $\rho_j$ such that $\pi_{\rho_j(k)\,|\,j} \geq \pi_{\rho_j(k+1)\,|\,j}$,*

$$\pi_k \propto \frac{1}{k^\alpha} \quad \text{and} \quad \pi_{\rho_j(k)\,|\,j} \propto \frac{1}{k^\alpha}, \quad \text{for all } j, k,$$

*where by $\pi_k \propto 1/k^\alpha$ we mean that the frequencies are normalized, $\pi_k = 1/zk^\alpha$ for $z = \sum_{k=1}^{d}1/k^\alpha$.*

This assumption may appear strong, as it would be satisfied for example if the words were sampled i.i.d. with frequencies $\pi_1, \ldots, \pi_d$ as $\pi_{k\,|\,j} = \pi_k$. But it does not require that all conditional distributions

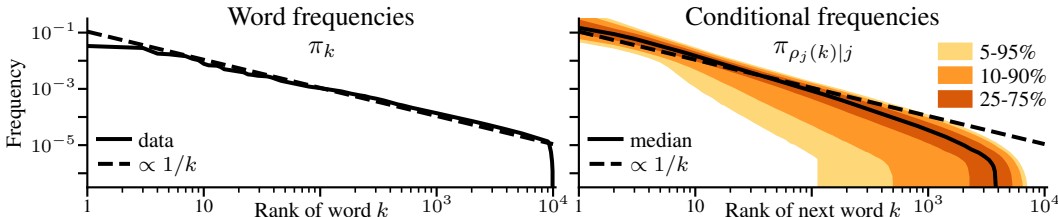

Figure 3: **Token frequencies and conditional frequencies approximately follow Zipf's law.** The approximation of Assumption 2.3 (− −) is a reasonable approximation of the frequencies (left) and conditional frequencies (right) on text data, computed on OpenWebText for a vocabulary of $10^4$ words. Right: median and quantiles of the next-word frequencies after sorting, $\pi_{\rho_j(k)\,|\,j}$ for $j \in [d]$.

be the same. The distribution of the next word after $j$ can depend on $j$. This assumption merely asks that, once sorted, the next-word frequencies follow a power law with the same exponent. Some distributions might deviate from this trend if a token can only logically be followed by specific tokens, or if the word being conditioned on is rare and our dataset is relatively small.[1] While we do not expect the assumption to be exactly satisfied in practice, it appears to be a reasonable high-level approximation of real-world data, as shown in Fig. 3 in comparison to the empirical distributions on OpenWebText, and leads to accurate predictions as shown in Fig. 2.

**Relation to other spectral conditions.** Even though Problem 2.1 is $d^2$-dimensional, the dynamics of GD are equivalent to those run on a $d$-dimensional problem as Proposition 2.2 can be rewritten as $\mathcal{L}_d(t) - \mathcal{L}_d^* = \frac{1}{2}\sum_{i=1}^d \pi_i(1-\eta\pi_i)^{2t}\Delta_i^2$ for $\Delta_i^2 = \sum_{j=1}^d \delta_{ij}(0)^2$. Many works have considered decay conditions on the eigenvalues and distances to the solution, $\pi_k \propto k^{-a}$ and $\Delta_k^2 \propto k^{-b}$, similar to the source/capacity conditions (Caponnetto and De Vito, 2007). However, their focus is typically on a fast decay, $a+b > 1$, which leads dimension-independent power-laws (see e.g., Paquette et al., 2024, and references therein). While Assumption 2.3 is a special case corresponding to $(a,b) = (\alpha, 0)$, we study the case $\alpha \leq 1$ to understand the behavior of optimizers on heavy-tailed data.

### 2.2 Strategy for the analysis

Our goal is to derive scaling laws for the loss of Problem 2.1 in $d$ dimensions after $t$ steps, $\mathcal{L}_d(t)$, as $d \to \infty$. Such scaling laws can be interpreted as approximating the convergence rate for large $d$, or serve as a guide on how to scale the hyperparameters of the optimizer as we increase the vocabulary size. Formally, we compute the asymptotic limit of the rate $r(t)$ at which the relative loss decreases,

$$\mathcal{L}_d(t) - \mathcal{L}_d^* \overset{d}{\sim} r(t)\big(\mathcal{L}_d(0) - \mathcal{L}_d^*\big), \quad \text{where } \overset{d}{\sim} \text{ is notation for} \quad \lim_{d\to\infty} \frac{\mathcal{L}_d(t) - \mathcal{L}_d^*}{\mathcal{L}_d(0) - \mathcal{L}_d^*} = r(t),$$

Works on scaling laws typically model the absolute value of the loss. This approach degenerates when the loss at initialization vanishes or diverges as $d \to \infty$ which happens when $\alpha \leq 1$. Considering the relative decrease circumvents the issue, as also noted by Bulatov (2023) and Tao et al. (2024).

Another potential degeneracy is the scaling of time. If the problem becomes more difficult as $d$ grows, it might be impossible to make progress in finite time. To take a concrete example, suppose that $\mathcal{L}_d^* = 0$ and $\mathcal{L}_d(t) = r_d(t)\mathcal{L}_d(0)$ with $r_d(t) = (1-1/d)^t$. If we take the limit as $d \to \infty$ for a fixed $t$, we obtain $\lim_{d\to\infty}(1-1/d)^t = 1$. The rate no longer depends on $t$, and we cannot make progress unless $t$ grows with $d$. If we instead introduce a rescaled time variable $\tau$ and scale $t_d(\tau) = \tau d$, we recover a linear rate in the rescaled time $\tau$ as $(1-1/d)^{\tau d} \overset{d}{\sim} e^{-\tau}$. A similar issue arises in random matrix theory, where the dimensions of the matrix are taken to grow jointly with a fixed ratio to avoid degenerate solutions (Potters and Bouchaud, 2020). It can be verified that $t_d(\tau) = \tau d$ is the "right" scaling, as the limit $r_d(t_d(\tau))$ degenerates otherwise. Using $f(x) \ll g(x)$ for $\lim_{x\to\infty} f(x)/g(x) = 0$, we have $r_d(t_d(\tau)) \overset{d}{\sim} 1$ if $t_d(\tau) \ll d$ and $r_d(t_d(\tau)) \overset{d}{\sim} 0$ if $t_d(\tau) \gg d$; we either make no progress or solve the problem instantly. Our results are derived by taking the finite dimensional rate $r_d(t)$ with a scaling $t_d$ such that the asymptotic rate $r(\tau)$ is well-defined in terms of the rescaled time $\tau$,

$$r(\tau) := \lim_{d\to\infty} r_d(t_d(\tau)) = \lim_{d\to\infty} \frac{\mathcal{L}_d(t_d(\tau)) - \mathcal{L}_d^*}{\mathcal{L}_d(0) - \mathcal{L}_d^*}. \tag{2}$$

---

[1]Even with i.i.d. data following $\pi_k \propto 1/k$, accurately estimating the conditional frequency takes many samples. With a vocabulary size of $d = 10^4$, we expect to see the pair $(x = d, y = d)$ once every $10^8$ tokens.

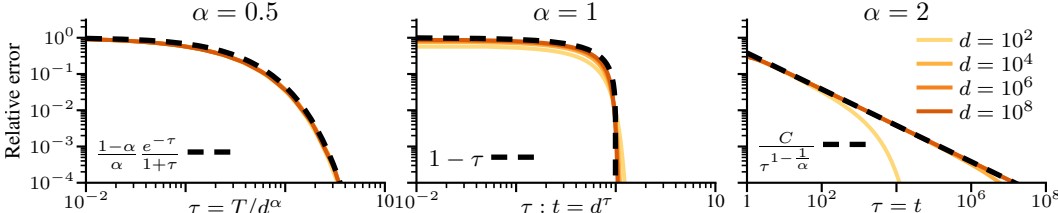

Figure 4: **Scaling of gradient descent on power-law data with exponent $\alpha$ (Theorem 3.1).** The dynamics of gradient descent on the linear bigram model with data satisfying Assumption 2.3 converge to our scaling law (- -, Theorem 3.1) as $d$ grows. Achieving a relative error $\varepsilon$ requires scaling the iteration budget $T$ with $d^\alpha$ for $\alpha < 1$, $T$ with $d^{1-\varepsilon}$ for $\alpha = 1$, and no scaling for $\alpha > 1$.

## 3 Scaling laws for gradient descent

We study the relative error of GD with the step-size $\eta = 1/\pi_1$, the inverse of the largest eigenvalue,

$$r_d(t) = \frac{\mathcal{L}_d(t) - \mathcal{L}_d^*}{\mathcal{L}_d(0) - \mathcal{L}_d^*} = \sum_{i=1}^{d} \pi_i \left(1 - \frac{\pi_i}{\pi_1}\right)^{2t}. \tag{3}$$

Before diving into the main result, we provide some intuition on those dynamics. The performance of GD depends on the speed of convergence for each word, $(1 - \pi_k/\pi_1)$, and the proportion of the error coming to that word, $\pi_k$. The parameter $\alpha$ controls both. If we increase $\alpha$, the frequencies $\pi_k \propto 1/k^\alpha$ decay faster. Low-frequency words converge more slowly, $(1 - \pi_k/\pi_1) = (1 - 1/k^\alpha)$, but contribute less to the error, $\pi_k = 1/zk^\alpha$ where the normalization term is $z = \sum_{k=1}^{d} 1/k^\alpha$.

**For $\alpha > 1$,** the error is dominated by high-frequency words, which converge quickly. The error attributed to the first $K$ words, $\sum_{k=1}^{K} \pi_k$, is a constant approximation of the total. Increasing the vocabulary size $d$ does not make the problem much harder as low-frequency words contribute little. **For $\alpha < 1$,** the error associated with the first $K$ words vanishes if $K$ is fixed and $d$ grows, indicating that most of the error comes from low-frequency words. However, their convergence speed improves as $\alpha$ decreases, with the extreme case of uniform frequencies at $\alpha = 0$, making the problem easier. **The case $\alpha = 1$** of Zipfian data exhibits the worst of both settings. The decay is slow enough that the contribution of low-frequency words is significant, but fast enough that their convergence is slow.

The following theorem formalizes these intuitions.

**Theorem 3.1** (Scaling for gradient descent). *On the bigram problem (Prob. 2.1) with distributions following a power law with exponent $\alpha > 0$ (Assumption 2.3), gradient descent with a step-size $1/\pi_1$, with time scaling $t_d(\tau)$ has the following asymptotic convergence rate (Eq. (2)).*

$$\text{If } \alpha < 1, \qquad t_d(\tau) = \tfrac{1}{2}\tau d^\alpha, \qquad r(\tau) = \frac{1-\alpha}{\alpha} E_{\frac{1}{\alpha}}(\tau) \backsim \frac{1-\alpha}{\alpha} \frac{e^{-\tau}}{\tau+1},$$

$$\text{if } \alpha = 1, \qquad t_d(\tau) = \tfrac{1}{2}d^\tau, \qquad r(\tau) = 1 - \tau \qquad \text{where } \tau \in [0,1],$$

$$\text{if } \alpha > 1, \qquad t_d(\tau) = \tau, \qquad r(\tau) \backsim \frac{B\left(1 - \frac{1}{\alpha}, 1 + 2t\right)}{\alpha\zeta(\alpha)} \backsim C\frac{1}{\tau^{1-\frac{1}{\alpha}}},$$

*where $\Gamma$ is the Gamma function, $E$ is the generalized exponential integral, $B$ is the Beta function, and $\zeta$ is the zeta function (DLMF, §5.2, §5.12 §8.19 §25.2), and $C = \Gamma\left(1 - \frac{1}{\alpha}\right)/\alpha\zeta(\alpha)$.*

*Proof.* We sketch the proof for $\alpha = 1$ and leave the remaining cases to Appendix C. Under Eq. (1) and Assumption 2.3 the dynamics of the normalized loss $r_d(t)$ (Eq. (3)) reduce to

$$r_d(t) = \frac{1}{H_{d,\alpha}} \sum_{k=1}^{d} k^{-\alpha} \left(1 - k^{-\alpha}\right)^{2t},$$

where $H_{d,\alpha} = \sum_{k=1}^{d} k^{-\alpha}$. To simplify the analysis, we use the integral form of the sum as we can use Laplace's method to estimate its behavior for large $d$, see Appendix C for a formal justification;

$$\text{For } \alpha = 1, \quad r_d(t) \approx I_d(t) := \frac{1}{H_{d,1}} \int_1^d k^{-1}\left(1 - k^{-1}\right)^{2t} dk = \frac{\log(d)}{H_{d,1}} \int_0^1 \left(1 - d^{-z}\right)^{2t} dz,$$

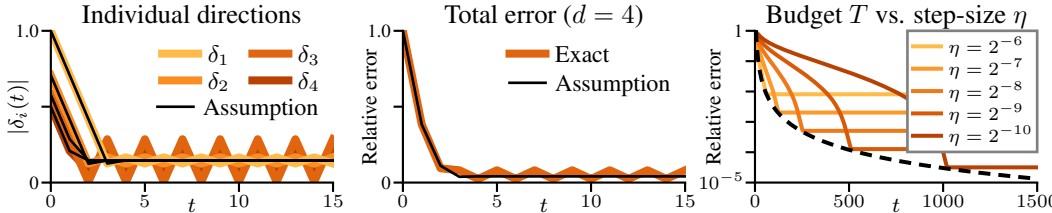

Figure 5: **Illustration of our modeling assumption for sign descent ([Assumption 4.1](#)).** Left: instead of modeling the oscillations of sign descent, we treat the oscillatory phase as constant. Middle: The effect on the total error. Right: Because SD eventually oscillates, the step-size needs to depend on the iteration budget $T$ to achieve best performance after $T$ steps (the envelope **– –**).

after the change of variable $k = d^z$ or $z = \log(k)/\log(d)$. As the normalizer $H_{d,1} \overset{d}{\sim} \log(d)$, we only need to consider the limit of the integral. Taking $d \to \infty$ with $t$ fixed, the integral converges to 1 and we make no progress, regardless of $t$. To make progress, $t$ needs to scale as $2t = d^\tau$ for $\tau \in [0, 1]$,

$$I_d(d^\tau) = \frac{\log(d)}{H_{d,\alpha}} \int_0^1 \left(1 - \frac{d^{\tau-z}}{d^\tau}\right)^{d^\tau} \mathrm{d}z.$$

For a fixed $\tau$ and as $d \to \infty$, the integrand converges to 0 if $z < \tau$ and 1 if $z > \tau$. As it is bounded by a constant, we can exchange limits and integrals by the dominated convergence theorem to obtain

$$\lim_{d \to \infty} \int_0^1 \left(1 - \frac{d^{\tau-z}}{d^\tau}\right)^{d^\tau} \mathrm{d}z = \int_0^\tau 0\mathrm{d}z + \int_\tau^1 1\mathrm{d}z = 1 - \tau. \qquad \square$$

The results highlight different regimes depending on $\alpha$. The number of iterations needs to scale with dimension if the data decays as slow as or slower than Zipf's law ($\alpha \leq 1$) whereas it is not necessary for lighter-tailed data ($\alpha > 1$). We show in [Fig. 4](#) that the dynamics on data satisfying [Assumption 2.3](#) converge to the asymptotic rates of [Theorem 3.1](#) and are accurate even for common vocabulary sizes.

## 4 Scaling laws for sign descent

The dynamics of SD differ qualitatively from those of GD as they take a uniform update in all directions, regardless of the magnitude of the derivatives. As we will see, this makes it better suited for the linear bigram model with Zipf-distributed data, but comes with additional challenges. For SD, we need to address two issues. First, the sign descent update is not linear; we need an alternative to the closed form solution of GD in [Eq. (1)](#). Second, SD does not converge with a fixed step-size; we need to scale step-size as a function of the iteration budget and dimension.

If run with a constant step-size, the update of sign descent with a step-size of $\eta$ is

$$\mathbf{W}_{t+1} = \mathbf{W}_t - \eta \operatorname{sign}(\nabla \mathcal{L}(\mathbf{W}_t)).$$

As the Hessian of [Problem 2.1](#) is diagonal, the update applies independently to each parameter. Letting $\delta_{ij}(t)$ be the distance along the $(i, j)$th parameter at step $t$,

$$\delta_{ij}(t+1) = \delta_{ij}(t) - \eta \operatorname{sign}(\delta_{ij}(t)).$$

The difficulty in the analysis comes from the fact that $|\delta_{ij}(t)|$ does not converge to 0. Instead, $|\delta_{ij}(t)|$ will oscillate between some $c \in (0, \eta)$ and $c - \eta$, unless $t = |\delta_{ij}(t)|/\eta$ is an integer and the distance to the solution reaches exactly 0. Keeping track of these oscillations is cumbersome, as each of the $d^2$ parameters will oscillate between different constants. To simplify the analysis, we assume that the distances decrease while $|\delta_{ij}(0)| \geq t\eta$ then go to $\eta/2$ to model the oscillatory regime, essentially "averaging" the oscillations, as illustrated in [Fig. 5](#).

**Assumption 4.1.** *We assume that sign descent with step-size $\eta$ follows the dynamics*

$$|\delta_{ij}(t)| := \begin{cases} |\delta_{ij}(0)| - t\eta & \text{if } |\delta_{ij}(0)| - t\eta \geq 0, \\ \eta/2 & \text{otherwise.} \end{cases}$$

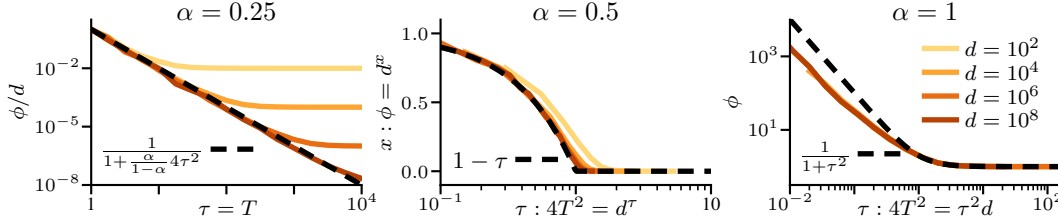

Figure 6: **Convergence of the best step-size for sign descent to the scaling in Definition 4.4.** The optimal step-size for $T$ steps of sign descent converge to our scaling (━ ━) given in Definition 4.4 (for $\tau > 1$ in the case of $\alpha = 1$). Computed by grid search on the linear bigram model with data satisfy Assumption 2.3.

Those dynamics do not capture the fact that a direction might reach exactly 0, after which sign descent would not oscillate, but this can only happen for a few directions if $|\delta_{ij}(0)| \propto 1/j^{\alpha}$, and their impact is small with large $d$. With this assumption, we have the following dynamics.

**Proposition 4.2.** *If the conditional distribution follows a power law with exponent $\alpha$ as in Assumption 2.3, the dynamics of sign descent with step-size $\eta$ in Assumption 4.1 lead to the loss*

$$\mathcal{L}_d(t,\eta) - \mathcal{L}_d^* := \sum_{i=1}^{d} \sum_{j=1}^{d} \lambda_{ij} \delta_{ij}(t)^2 = \sum_{k=1}^{k_*} (\delta_k(0) - t\eta)^2 + \sum_{k=k_*+1}^{d} \left(\frac{\eta}{2}\right)^2 \quad where \quad \delta_k(0) = \pi_k,$$

*and $k_*$ is the number of directions in the decreasing regime, $k_* = \max_k k : \pi_k > t\eta$*

*Proof.* By Proposition 2.2, $\lambda_{ij} = \pi_i$ does not depend on $j$. By Assumption 2.3, there is a permutation $\rho_i$ such that $\delta_{i\rho_i(j)}(0) = \pi_j$. As a result, the dynamics of $\delta_{i\rho_i(j)}(t)$ do not depend on $i$. Writing $\delta_j(t)$ as a shortcut for $\delta_{i,\rho_i(j)}(t)$ for any $i$ and using that $\sum_{i=1}^{d} \pi_i = 1$,

$$\sum_{i=1}^{d} \sum_{j=1}^{d} \lambda_{ij} \delta_{ij}(t)^2 = \sum_{i=1}^{d} \pi_i \sum_{j=1}^{d} \delta_{ij}(t)^2 = \sum_{i=1}^{d} \pi_i \sum_{j=1}^{d} \delta_j(t)^2 = \sum_{j=1}^{d} \delta_j(t)^2.$$

We then split the sum depending on whether $|\delta_k(t)|$ is decreasing or oscillating. $\qquad\square$

The dynamics of SD in Proposition 4.2, differ qualitatively from those of GD in Eq. (1). The progress in each direction is not scaled by $\pi_i$, because the update is uniform across directions. This is what will enable SD to make faster progress on low-frequency words. However, we now have another challenge in that we need to choose the step-size $\eta$ to trade-off between the oscillations of magnitude $(\eta/2)^2$ on low-frequency words and still making progress on high-frequency words that are not yet in the oscillatory regime. This is easy when $\alpha$ is small and the frequencies are close to uniform, giving a small spread for the initial distances $\delta_k(0)$, but becomes more difficult as $\alpha$ increases. From this, we expect SD to perform better than GD for small $\alpha$, and worse for large $\alpha$, but we need to understand how to set the step-size to understand where the transition happens.

### 4.1 Scaling of the step-size

As SD with a fixed step-size eventually enters an oscillatory regime, the loss we converge to as $t$ grows depends on $\eta$. To describe the performance achievable after tuning $\eta$ for a given budget $T$, we need to estimate how $\eta$ scales with $T$ and $d$. This effect is illustrated in Fig. 5 (right). We use capital $T$ to emphasize that we are modeling the loss at the end of a training run of $T$ steps with a fixed step-size which depends on $T$. Getting the exact form of $\eta_* = \arg\min_\eta \mathcal{L}_d(T,\eta)$ is out of reach, but we establish bounds on the optimal step-size.

**Proposition 4.3.** *The step-size $\eta_*$ that $\mathcal{L}_d(T,\eta)$ in Proposition 4.2 given $T$ and $d$, satisfies*

$$\frac{\delta_d(0)}{T} \le \eta_* \le \frac{\delta_1(0)}{T}.$$

*Proof.* If $\eta \le \delta_d(0)/T$, all directions are still in the decreasing regime of Assumption 4.1 at time $T$. As long as $T\eta < \delta_d(0)$, increasing the step-size leads to more progress. Similarly, if $T\eta \ge \delta_1(0)$, all directions are in the oscillatory regime, and reducing the step-size reduces the oscillations. $\qquad\square$

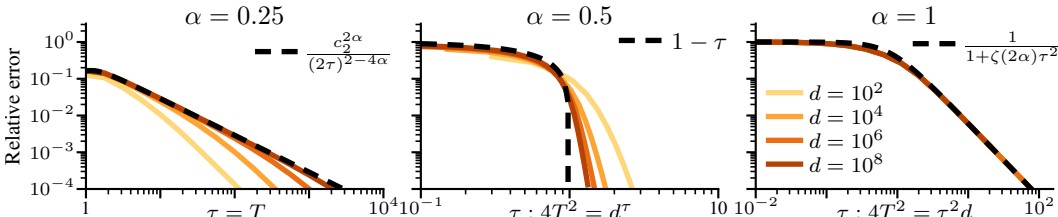

Figure 7: **Scaling of sign descent on power-law data with exponent** $\alpha$ (**Theorem 4.5**). The dynamics of sign descent on the linear bigram model with data satisfying Assumption 2.3 converge to our scaling law (- -) as $d$ grows, as described in Theorem 4.5. Achieving a relative error $\varepsilon$ requires no scaling for $\alpha < 1/2$, scaling $t$ with $d^{(1-\varepsilon)/2}$ for $\alpha = 1/2$, and $t$ with $d^{1/2}$ for $\alpha > 1/2$.

As our initial distances follow a power law, $\delta_k(0) = \pi_k = \frac{1}{zk^\alpha}$ where $z = \sum_{k=1}^d k^{-\alpha}$, Proposition 4.3 suggests an alternative parameterization of the step-size as

$$\eta(\phi) = \frac{1}{zT\phi^\alpha} \quad \text{with} \quad 1 \le \phi \le d,$$

where $\phi$ controls how many directions are still decreasing. We now define the following scaling of $\phi$.

**Definition 4.4.** *We define the following scalings as a function of the dimension $d$ and rescaled time $\tau$*

$$\text{if } \alpha < 1/2, \quad T_d(\tau) = \tau, \qquad \phi_d(\tau) = \begin{cases} d & \text{if } c_1 + 4c_2\tau^2 \le 1, \\ d\big(c_1 + 4c_2\tau^2\big)^{-1} & \text{otherwise}, \end{cases}$$

$$\text{if } \alpha = 1/2, \quad T_d(\tau) = \tfrac{1}{2}d^{\tau/2}, \quad \phi_d(\tau) = d^{1-\tau}, \quad \text{where } \tau \in [0,1],$$

$$\text{if } \alpha > 1/2, \quad T_d(\tau) = \tfrac{1}{2}\tau\sqrt{d}, \quad \phi_d(\tau) = \begin{cases} 1 + 1/\tau^2 & \text{if } \tau^2 < (2^\alpha - 1)^{-1} \text{ and } \alpha < 1, \\ (1 + 1/\tau^2)^{1/\alpha} & \text{otherwise}, \end{cases}$$

*where $c_1 = 1 - \frac{1}{2\alpha}$, $c_2 = \frac{\alpha}{1-\alpha}$.*

While those scalings need not be optimal, they match the empirical behavior of the best step-size computed by grid-search, as shown in Fig. 6. For $\alpha > 1/2$, the step-size is only accurate for $\tau^2 \ge 1/(2^\alpha - 1)$ or $\tau \ge 1$ for $\alpha = 1$. We justify those estimates in Appendix D.

### 4.2 Asymptotic behavior

Using the scalings for $T$ and $\phi$ in Definition 4.4, we define the asymptotic rate of sign descent as

$$r(\tau) = \lim_{d\to\infty} \frac{\mathcal{L}_d(T_d(\tau), \phi_d(\tau)) - \mathcal{L}_d^*}{\mathcal{L}_d(0) - \mathcal{L}_d^*}. \tag{4}$$

**Theorem 4.5** (Scaling for sign descent). *Given scalings for $T$ and $\phi$ in Definition 4.4, the asymptotic convergence rate of sign descent (Eq. (4)) is, with $c_1 = 1 - \frac{1}{2\alpha}$, $c_2 = \frac{\alpha}{1-\alpha}$,*

$$\text{if } \alpha < 1/2, \quad T_d(\tau) = \tau, \qquad r(\tau) = \begin{cases} 2\alpha c_2 & \text{if } \tau^2 \le \frac{1-c_1}{4c_2} \\ \frac{(c_1 + c_2 4\tau^2)^{2\alpha}}{4\tau^2} & \text{otherwise} \end{cases} \underset{\sim}{} \frac{c_2^{2\alpha}}{(2\tau)^{2-4\alpha}},$$

$$\text{if } \alpha = 1/2, \quad T_d(\tau) = \tfrac{1}{2}d^{\frac{1}{2}\tau}, \qquad r(\tau) = 1 - \tau, \quad \text{where } \tau \in [0,1],$$

$$\text{if } \alpha > 1/2, \quad T_d(\tau) = \tfrac{1}{2}\tau\sqrt{d}, \qquad r(\tau) \underset{\sim}{} \frac{1}{1 + \zeta(2\alpha)\tau^2}.$$

We leave the proofs in Appendix D. The results also show different forms of scaling depending on $\alpha$, with a threshold at $\alpha = 1/2$ instead of 1. The scaling in dimension is flipped compared to GD. SD needs $t$ to scale with $d$ when $\alpha$ is large, which is the regime where GD can make progress with finite $t$. For the case of Zipf-distributed data ($\alpha = 1$), SD only needs a scaling in $d^{1/2}$ compared to the $d^{1-\varepsilon}$ scaling of GD, showing that it achieves better performance for $\varepsilon < 1/2$. We show in Fig. 7 that the asymptotic rates of Theorem 3.1 are accurate even for finite $d$.

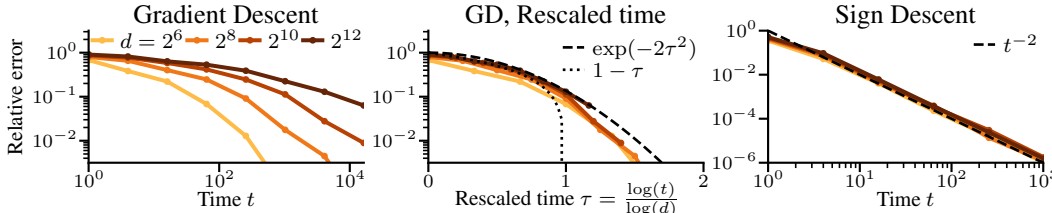

Figure 8: **The difference in scaling with dimension also occurs with the cross-entropy loss.** Relative error on a linear bigram problem with cross-entropy loss with vocabulary size $d$ when the word frequencies follow Zipf's law, $\pi_k \propto 1/k$. For GD (left), the performance depends heavily on the dimension. Its performance is similar to the scaling of $1 - \tau$ for $t = d^\tau$ found for the square loss while $\tau < 1$, and worse for $\tau > 1$ (middle). The performance of SD appears independent of $d$ (right).

## 5    Conclusion

We have presented scaling laws for gradient descent (GD) and sign descent (SD) on the linear bigram model as a function of the power law exponent $\alpha$ of the word frequencies. Rather than hide the dimension dependence in problem specific constants, we consider the scaling of running time and dimension as the problem grows in size to get precise estimates of the scaling. Our results highlight the benefit of SD and the need to address ill-conditioning to improve the performance of GD.

Our results show that the power-law scaling is specific to the regime $\alpha > 1$. This regime may accurately describe cases where the training dynamics converge to a well-defined limit, such as when increasing width or depth (Yang et al., 2021; Bordelon et al., 2024b; Noci et al., 2024), it misses a large dimension dependence as we scale the vocabulary size. The scaling we obtain for $\alpha \leq 1$ have a different functional form and highlight the dependency on dimension. For GD on Zipf-distributed data, the scaling of $d^{1-\varepsilon}$ shows a non-trivial interplay between the desired error $\varepsilon$ and the dimension. Our results suggest that increasing the vocabulary size might require a larger training budget, not only because each iteration is more costly due to the larger embedding matrices, but also because more iterations are needed to reach the same error. Algorithms that target this dimension dependence, for example by estimating word frequencies (Li et al., 2022), would be an interesting next step.

Our approach however has limitations. We do not cover the online case, for which the analysis should be extendable using existing tools. The addition of momentum for sign descent would be more complex but particularly interesting to dampen oscillations, and getting finite-dimensional results by tracking a correction term for finite $d$ would be enlightening, as the convergence to the asymptotic regime can sometimes be slow, especially in the case $\alpha = 1$. A more difficult extension would be to consider models leading to non-linear dynamics, such as bilinear models (Mikolov et al., 2013) or the cross-entropy loss. But it is not clear how to obtain closed-form solutions or sufficiently accurate approximations even for GD in the deterministic setting. We can however probe the behavior of GD and SD experimentally, and present preliminary results with the cross-entropy loss.

**Empirical behavior with cross-entropy loss.** We experiment with a variant of the linear bigram model trained with cross-entropy loss on synthetic data satisfying Assumption 2.3 for $\alpha = 1$. We show in Fig. 8 the result of training models with increasing vocabulary, with the step-size set by grid-search for both GD and SD to minimize the loss at the given horizon. The results suggest that the gap in scaling between GD and SD is even larger than with the quadratic loss; GD appears to require $t \sim d$, as in the quadratic case, while the performance of SD appears independent of $d$.

## Acknowledgments and Disclosure of Funding

We thank Si Yi (Cathy) Meng, Aaron Mishkin, and Victor Sanches Portella for helpful discussions and providing comments on the manuscript, and for the feedback from anonymous reviewers. This work has received support from the French government, managed by the National Research Agency, under the France 2030 program with the reference "PR[AI]RIE-PSAI" (ANR-23-IACL-0008). Frederik Kunstner is supported by a Marie Skłodowska-Curie Fellowship from the European Union's Horizon Europe Research and Innovation program under Grant Agreement No. 101210427.

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

# Supplementary Material

The supplementary material is organized as follows.

- Appendix A gives experimental details and information on how to reproduce the figures.

- Appendix B compares our results to standard convergence rates in the literature.

- Appendix C gives the main results for gradient descent Theorem 3.1.

- Appendix D gives the main results for sign descent Theorem 4.5.

## A    Experimental details

This section goes over the technical details of the experiments needed to reproduce the figures.

### A.1    Computational complexity

We use $d$ to denote the size of the vocabulary, but the number of parameters $\mathbf{W}$ is $d^2$ as we have to learn the conditional probability table $\pi_{k\,|\,j}$. As the number of iterations $t$ has to scale with dimension, the problem scales in $d^3$, which becomes prohibitive fast. To circumvent this issue, we use the fact that the training dynamics of gradient descent and sign descent on data following Assumption 4.1 can be simulated in $O(d)$. The error after $t$ iterations can then be computed in closed-form if initialized at $0$, making it possible to compute the loss after $t$ steps without computing the intermediate steps.

**Proposition A.1** (Reduction of the dynamics for gradient descent). *Under Assumption 2.3, the dynamics of gradient descent with step-size $1/\pi_1$ can be computed in $O(d)$ as*

$$r_d(t) := \frac{\mathcal{L}_d(t) - \mathcal{L}_d^*}{\mathcal{L}_d(0) - \mathcal{L}_d^*} = \frac{1}{\sum_{k=1}^d k^{-\alpha}} \sum_{k=1}^d \frac{1}{k^\alpha}\left(1 - \frac{1}{k^\alpha}\right)^{2t}.$$

*Proof.* We use the dynamics using the eigendecomposition notation presented in Section 2,

$$r_d(t) = \mathcal{L}_d(t) - \mathcal{L}_d^* = \sum_{i=1}^d \sum_{j=1}^d \lambda_{ij}\delta_{ij}(t)^2, \qquad \text{and} \qquad \delta_{ij}(t) == (1 - \lambda_{ij})^t \delta_{ij}(0).$$

Using Assumption 2.3 gives that $\lambda_{ij}$ is independent of $j$ and $\delta_{ij}$ is independent of $i$ as

$$\lambda_{ij} = \pi_i = \frac{1}{zi^\alpha} \qquad \delta_{ij}(0) = \pi_{\rho_i(j),i} = \frac{1}{zj^\alpha} \qquad \text{where} \qquad z = \sum_{k=1}^d \frac{1}{k^\alpha}.$$

Plugging in those together and using that the step-size is $\eta = \pi_1 = 1/z$ gives

$$\frac{\mathcal{L}_d(t) - \mathcal{L}_d^*}{\mathcal{L}_d(0) - \mathcal{L}_d^*} = \frac{\sum_{i=1}^d \sum_{j=1}^d \frac{1}{zi^\alpha}\left(1 - \frac{1}{i^\alpha}\right)^{2t}\delta_{ij}(0)^2}{\sum_{i=1}^d \sum_{j=1}^d \frac{1}{zi^\alpha}\delta_{ij}(0)^2},$$

$$= \frac{\sum_{i=1}^d \frac{1}{i^\alpha}\left(1 - \frac{1}{i^\alpha}\right)^{2t}\sum_{j=1}^d \delta_{ij}(0)^2}{\sum_{i=1}^d \frac{1}{i^\alpha}\sum_{j=1}^d \lambda_{ij}\delta_{ij}(0)^2} = \frac{\sum_{i=1}^d \frac{1}{i^\alpha}\left(1 - \frac{1}{i^\alpha}\right)^{2t}}{\sum_{i=1}^d \frac{1}{i^\alpha}}. \qquad \square$$

**Proposition A.2** (Reduction of the dynamics for sign descent). *Under Assumption 2.3, the simplified dynamics of sign descent (Assumption 4.1) with step-size $\eta(T,\phi) = 1/zT\phi^\alpha$ following the reparameterization of Proposition 4.3 where $z = \sum_{k=1}^d k^{-\alpha}$ can be computed in $O(d)$ as*

$$r_d(T,\phi) := \frac{\mathcal{L}_d(T, \eta(T,\phi)) - \mathcal{L}_d^*}{\mathcal{L}_d(0) - \mathcal{L}_d^*} = \frac{1}{\sum_{k=1}^d k^{2\alpha}} \sum_{k=1}^d \left( \begin{cases} \frac{1}{k^\alpha} - \frac{1}{\phi^\alpha} & \text{if } |\delta_{ij}(T-1)| - \eta \geq 0, \\ \frac{1}{2\phi^\alpha} & \text{otherwise}, \end{cases} \right)^2,$$

*Proof.* Using the same derivation as above for Proposition A.1 but using the update dynamics assumed in Assumption 4.1. Note that those dynamics imply that $\delta_{ij}(T)$ is independent of $i$. Writing $\Delta_j =$

$\delta_{ij}(T)$ for any $i$ and using that $\sum_{i=1}^{d} \pi_i = 1$, we have

$$\frac{\mathcal{L}_d(T) - \mathcal{L}_d^*}{\mathcal{L}_d(0) - \mathcal{L}_d^*} = \frac{\sum_{i=1}^{d} \sum_{j=1}^{d} \lambda_{ij} \delta_{ij}(T)^2}{\sum_{i=1}^{d} \sum_{j=1}^{d} \lambda_{ij} \delta_{ij}(0)^2} = \frac{\sum_{i=1}^{d} \sum_{j=1}^{d} \pi_i \Delta_j(T)^2}{\sum_{i=1}^{d} \sum_{j=1}^{d} \pi_i \Delta_j(0)^2} = \frac{\sum_{j=1}^{d} \Delta_j(T)^2}{\sum_{j=1}^{d} \Delta_j(0)^2}.$$

Expanding $\Delta_j(T)$ using Assumption 4.1 gives the result. $\qquad\square$

For the real data experiments in Fig. 2, computing the dynamics cannot be reduced to $O(d)$. We still use the fact that the dynamics can be computed in closed-form to avoid running $t$ steps of gradient/sign descent. For sign descent, we do not use the simpler model of Assumption 2.3 but the full dynamics by computing the point reached after $t$ steps, including oscillations.

**Proposition A.3.** *Under the dynamics of sign descent with step-size $\eta$,*

$$\delta_{ij}(t+1) = \delta_{ij}(t) - \eta \operatorname{sign}(\delta_{ij}(t)),$$

*the distance after $t$ steps is given by*

$$\delta_{ij}(t) = \begin{cases} \delta_{ij}(0) - \eta t & \text{if } t \leq T_{\text{switch}}, \\ c_{ij} & \text{if } t - T_{\text{switch}} \text{ is odd}, \\ c_{ij} - \eta & \text{if } t - T_{\text{switch}} \text{ is even}, \end{cases} \quad \text{where} \quad \begin{aligned} T_{\text{switch}} &= \lfloor \delta_{ij}(0)/\eta \rfloor, \\ c_{ij} &= \delta_{ij}(0) - T_{\text{switch}}\eta. \end{aligned}$$

## A.2 Additional details about the figures

Fig. 1 shows the dynamics of gradient descent on Problem 2.1 on data satisfying Assumption 2.3.

Fig. 2 shows the dynamics on real data on the OpenWebText dataset (Gokaslan et al., 2019). Using the SentencePiece (Kudo and Richardson, 2018) implementation of BPE Sennrich et al., 2016, we train tokenizers with vocabulary sizes of $1\,000$, $3\,612$, $10\,000$ and $31\,622$ tokens on the first $2\,000\,000$ entries of the dataset with a maximum sentence length of $16\,768$. We compute the frequencies and conditional frequency tables for each vocabulary size using the entire dataset. We use the closed form formulas for the loss after $t$ steps using $O(d^2)$ computation detailed in the previous section to avoid having to run gradient and sign descent on those large models.

Gradient descent uses the empirically-derived step-size of $1/\pi_1$. For sign descent, for a given time horizon $T$, we optimize over the step-size numerically. Because the loss after $T$ steps as a function of the step-size is unimodal, we use the default bounded bracketing method in scipy (Virtanen et al., 2020, `minimize_scalar`) starting with the interval $[\eta_{\min}/d, d\eta_{\max}]$ where $\eta_{\min}, \eta_{\max}$ are the bounds derived in Proposition 4.3. The optimal step-size can vary drastically if it is computed on even or odd iterations as the loss oscillates. To avoid this issue, we only show even iterations.

Fig. 3 shows the frequencies computed as for Fig. 2 for the largest vocabulary size, $d = 31\,622$.

**The rightmost plot of Fig. 5** shows the simplified dynamics of sign descent.

Fig. 4, Fig. 6 and Fig. 7 show the convergence of the loss in $d$ dimension computed using the equations in Appendix A.1. For sign descent, the best step-size is obtained by grid search. We know the optimal step-size satisfies $\phi \in [1, d]$ (Proposition 4.3), so let $\phi = d^x$ where $x$ comes from a logarithmically spaced grid-search on $x$ from $-10$ to $0$, taking every $1/32$th powers;

$$\phi \in \{d^x : x \in \{10^{-10}, 10^{-10+\frac{1}{32}}, 10^{-10+\frac{2}{32}}, \ldots, 10^0\}\}.$$

Fig. 8 shows the dynamics of GD and SD on the linear bigram problem trained with the cross-entropy loss. For both GD and SD, the step-size is selected by grid-search with a similar $1/32$th power logarithmic grid as above, to minize the loss after $t$ steps. As for the plots of SD, Fig. 8 does not not show a single run but the envelope of the performance achievable with a constant step-size for $T$ steps. We have not found a way to simplify the computational complexity of the experiments using the cross-entropy loss. Each run requires running GD or SD for $t$ steps on the full $d \times d$ matrix. As $t$ needs to scale with $d$, computing a run of GD or SD takes $O(d^3)$ time, which limits the vocabulary sizes we can consider.

# B Comparison with worst-case rates

In this section, we compare our rates against results obtained using classical analyses to highlight the benefit of the asymptotic analysis in capturing the dependence on dimension. Our goal is not to imply those bounds are poor; each of the work cited below studied a specific problem and the assumptions were selected to highlight the impact of the condition number, non-convexity, variance, or other issue. However, due to their worst-case generality, existing results do not capture the dimension dependence on the problem of the linear bigram problem (Problem 2.1) with Zipf-distributed frequencies (Assumption 2.3) and predict worse behavior than actually observed.

In this section, we focus on Zipf-distributed data ($\alpha = 1$) as it is the most relevant to text data. To simplify notation, we assume that the conditional frequencies directly follow a power-law $\pi_{k\,|\,i} \propto 1/k$, instead of assuming that there exists a reordering $\rho_i$ such that $\pi_{\rho_i(k)\,|\,i} \propto 1/k$ as in Assumption 2.3. This reordering does not affect the dynamics of the loss and can be ignored without loss of generality.

## B.1 Standard smooth, (strongly-)convex rates.

Classical results in smooth, convex optimization are derived under the assumption that the objective function $\mathcal{L}_d$ is $L$-smooth and $\mu$-strongly convex with $\mu \geq 0$. We write the function rates in matrix form for the loss $\mathcal{L}_d$ defined in Problem 2.1, but this could equivalently be transformed to a vector form using and $\|\mathbf{x} - \mathbf{x}_*\|_2^2 = \|\mathbf{W} - \mathbf{W}_*\|^2$ if $\mathbf{x} = \text{vec}(\mathbf{W})$ and $\mathbf{x}_* = \text{vec}(\mathbf{W}_*)$ where vec stacks the columns of $\mathbf{W}$ as a single vector. For a twice-differentiable function, this is equivalent to assuming that the eigenvalues of the Hessian are bounded by $\mu \leq \lambda_{ij} \leq L$ for all $i, j \in [d]$ at every possible input. We compare against simple forms available in this setting (Nesterov (2018, Cor. 2.1.2), Boyd and Vandenberghe (2004, Eq. 9.18)). While it is possible to slightly improve the constants in these bounds, these constants do not meaningfully affect the asymptotic behavior as $d$ grows.

$$\mathcal{L}_d(t) - \mathcal{L}_d^* \leq \frac{2L\|\mathbf{W}_0 - \mathbf{W}_*\|_F^2}{t}, \qquad \mathcal{L}_d(t) - \mathcal{L}_d^* \leq \left(1 - \frac{\mu}{L}\right)^t (\mathcal{L}_d(0) - \mathcal{L}_d^*).$$

To better compare these rates with our results, we normalize them by $\mathcal{L}_d(0) - \mathcal{L}_d^*$,

$$\frac{\mathcal{L}_d(t) - \mathcal{L}_d^*}{\mathcal{L}_d(0) - \mathcal{L}_d^*} \leq \frac{L\|\mathbf{W}_0 - \mathbf{W}_*\|_F^2}{t(\mathcal{L}_d(0) - \mathcal{L}_d^*)} =: r_d^{\text{sub}}(t), \qquad \frac{\mathcal{L}_d(t) - \mathcal{L}_d^*}{\mathcal{L}_d(0) - \mathcal{L}_d^*} \leq \left(1 - \frac{\mu}{L}\right)^t =: r_d^{\text{lin}}(t).$$

**Proposition B.1** (Values of the constants). *On Problem 2.1 with frequencies following a power-law with $\alpha = 1$ (Assumption 2.3) initialized at $\mathbf{W}_0 = 0$, the smooth convex sublinear rate $r_d^{\text{sub}}(t)$ and the smooth strongly-convex linear rate $r_d^{\text{lin}}(t)$ are asymptotically equivalent to*

$$r_d^{\text{sub}}(t) \overset{d}{\sim} 2\frac{d}{\log(d)}\frac{1}{t}, \qquad r_d^{\text{lin}}(t) \overset{d}{\sim} \left(1 - \frac{1}{d}\right)^t.$$

*Proof.* The proof follow from substituting the constants with the values

$$\mu = \frac{1}{dz}, \qquad L = \frac{1}{z}, \qquad \|\mathbf{W}_0 - \mathbf{W}^*\|_F^2 = d(\mathcal{L}_d(\mathbf{W}_0) - \mathcal{L}_d^*).$$

where $z = \sum_{k=1}^d 1/k \overset{d}{\sim} \log(d)$. The eigenvalues are $\lambda_{ij} = \pi_i = 1/zi$ after normalization, giving $L = 1/z$ and $\mu = 1/zd$. Using that $\delta_{ij}(0) = 1/zj$ gives the loss and distance at initialization,

$$\mathcal{L}_d(\mathbf{W}_0) - \mathcal{L}_d^* = \sum_{i=1}^d \sum_{j=1}^d \lambda_{ij}\delta_{ij}(0)^2 = \sum_{i=1}^d \pi_i \sum_{j=1}^d \pi_{j\,|\,i}^2 = \sum_{j=1}^d \left(\frac{1}{zj}\right)^2,$$

$$\|\mathbf{W}_0 - \mathbf{W}_*\|^2 = \sum_{i=1}^d \sum_{j=1}^d \delta_{ij}(0)^2 = d\sum_{j=1}^d \left(\frac{1}{zj}\right)^2 = d(\mathcal{L}_d(\mathbf{W}_0) - \mathcal{L}_d^*). \qquad \square$$

Both rates struggle to predict the progress in "early" iterations, when $t$ is much smaller than $d$. The sublinear rate requires a scaling $t \propto d/\log(d)$ while the linear rate predicts $t \propto d$. Neither captures the progress that can be made by running $t = d^{1/2}$ iterations, which reaches an error of $\varepsilon = 1/2$. Instead, both rates predict no progress. We visualize the given rates in Fig. 9 after rescaling the number of

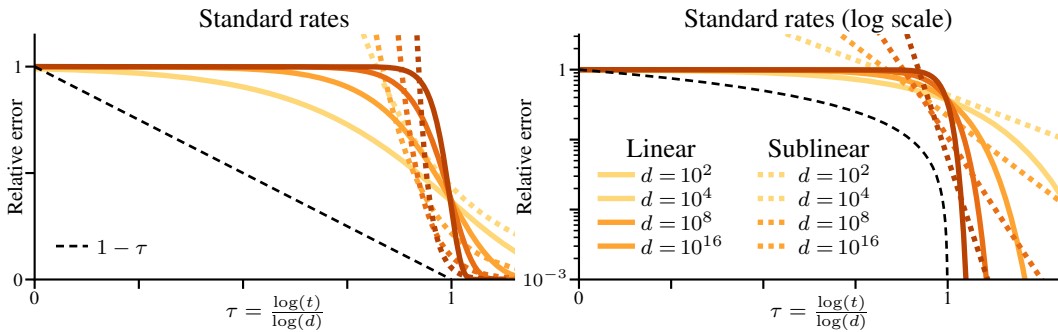

Figure 9: **Standard convergence rates do not capture the scaling in dimension.** Comparison of the standard linear and sublinear rates obtained for GD on the linear bigram model with Zipf-distributed data ($\alpha = 1$) with our asymptotic rate. The sublinear rate predicts a relative error greater than 1 until $t \approx d$, and the linear rate only reach the error $1/e$ at $t = d$. Our rate captures the fact that GD makes progress even with $t < d$.

steps to our normalized time $\tau = \log(t)/\log(d)$. The linear and sublinear rates are not converging to $r(\tau) = 1 - \tau$. Instead, they exhibit a sharper and sharper transition between not predicting any progress for $\tau < 1$ ($r(\tau) \approx 1$ or $r(\tau) > 1$) and that the problem is solved if $\tau > 1$.

## B.2 Rates for sign descent

Analyses on sign-like methods in the literature typically target more complex algorithms such as RMSProp (Tieleman and Hinton, 2012) or AdaGrad (Duchi et al., 2011) for Das et al. (2024) and Liu et al. (2025), or consider more general problems including non-convex functions for Bernstein et al. (2018) and Safaryan and Richtárik (2021). We are not aware of existing analyses that specifically target sign descent on diagonal quadratic problems such as Problem 2.1. This makes a direct comparison difficult. It might be that the rates described in those papers for the chosen problem setting or algorithm are tight. However, our message is that the resulting rates are too pessimistic even for a problem as simple as Problem 2.1 and suggest runtimes for sign descent that are off by a factor depending on the dimension.

The main difficulty in studying sign descent and sign-like methods more generally is the strong dependence on the coordinate system used. For Problem 2.1 the dynamics perfectly separate along coordinates which makes it possible to derive a closed form for the dynamics. Other works typically rely on assumptions on the Hessian that quantify how close to diagonal it is. For example, bound the Hessian with a diagonal matrix $\mathbf{L}$, $\mathbf{H} \preceq \mathbf{L}$ in Loewner ordering, and obtain rates that depend on the trace of $\mathbf{L}$ (e.g., Bernstein et al., 2018; Liu et al., 2025). For Problem 2.1, the Hessian is diagonal and made of $d$ diagonal copies of $\mathbf{X}^\top \mathbf{X}/n = \text{Diag}([\pi_1, ..., \pi_d])$, thus $\text{Tr}(\mathbf{L}) = \text{Tr}(\nabla^2 \mathcal{L}_d(\mathbf{W})) = d$.

**Anisotropic smoothness and AdaGrad.** Using this assumption, Liu et al. (2020, Theorem 4.1) show the following convergence rate for AdaGrad. To simplify their results and show the rate in its best light, we assume there is no noise in the gradient ($\|\boldsymbol{\sigma}\|_1 = 0$ in their notation), that AdaGrad is run with the parameter $\epsilon = 0$, that the algorithm is run with projections onto the constrained set $\mathcal{W} = \{\mathbf{W} : \|\mathbf{W}\|_\infty \leq \pi_1\}$ and that we initialize at $\mathbf{W} = 0$.

$$\mathcal{L}_d(t) - \mathcal{L}_d^* \leq \frac{\text{Tr}(\mathbf{L})\pi_1}{T}.$$

Normalizing the loss and simplifying the constants using the same approach as in Proposition B.1 gives the following asymptotic upper bound

$$\frac{\mathcal{L}_d(t) - \mathcal{L}_d^*}{\mathcal{L}_d(0) - \mathcal{L}_d^*} \leq r_d^{\text{Adagrad}}(t) := \frac{\text{Tr}(\mathbf{L})\pi_1}{T(\mathcal{L}_d(0) - \mathcal{L}_d^*)} \overset{d}{\sim} \frac{d\log(d)}{T} \frac{6}{\pi^2}.$$

Although we might expect Adagrad to outperform sign descent as it uses decreasing step-sizes to avoid the oscillations, this rate estimate that the number of iterations should scale with $d\log(d)$ instead of the scaling of $d^{1/2}$ we find for sign descent.

**Preconditioning effect of Adam.** Das et al. (2024) study RMSProp, or Adam without momentum ($\beta_1 = 0$) but with momentum on the moving average of the squared gradient. They use high-probability arguments to handle the dynamics of the preconditioner and random initialization. Their rate shows that Adam can perform better on diagonal quadratics if the condition number scales worse than linearly with the dimensionality, by replacing the condition number $\kappa$ with $\kappa_{\text{Adam}} = \min\{d_{\mathbf{W}} + 1, \kappa\}$ where $d_{\mathbf{W}}$ is the dimensionality of $\mathbf{W}$. Assuming that their bound holds with probability 1 with $\mathbf{W}_0 = 0$ and ignoring logarithmic factors in $d$ and $\epsilon$, their rate for diagonal quadratics is (Das et al., 2024, Thm. 2)

$$\mathcal{L}_d(t) - \mathcal{L}_d^* \leq \frac{\epsilon^2}{2}, \qquad \text{after} \qquad t \geq \tilde{O}(\kappa_{\text{Adam}}).$$

Unfortunately, on Problem 2.1 the dimensionality is $d_{\mathbf{W}} = d^2$ while the condition number scales as $\kappa = d$ with Zipfian eigenvalues ($\alpha = 1$) so the proposed approach does not improve over gradient descent. Normalizing the loss and using the same approach as in Proposition B.1 gives

$$\frac{\mathcal{L}_d(t) - \mathcal{L}_d^*}{\mathcal{L}_d(0) - \mathcal{L}_d^*} \leq \frac{\epsilon^2}{2}, \qquad \text{after} \qquad t \geq \tilde{O}(d).$$

This scaling predicts the same performance for Adam and gradient descent (up to log factors depending on $d$ and $\epsilon$ that we ignored) whereas our analysis shows a scaling of $d^{1/2}$ for sign descent.

**Non-convex results.** Results in the non-convex setting (Bernstein et al., 2018; Balles et al., 2020; Safaryan and Richtárik, 2021; Liu et al., 2025) give convergence results to stationarity instead of convergence in optimality gap, measured using the 1-norm of the gradient instead of the Euclidean norm. Because $\|\mathbf{v}\|_1^2 \leq \|\mathbf{v}\|_2^2 d$ for a $d$-dimensional vector $\mathbf{v}$, the time required to get the 1-norm small might be much worse than the time required to find a stationary point in Euclidean norm or to minimize the function value. To illustrate this point, we show that it is possible to have arbitrarily small relative error on Problem 2.1 and arbitrarily large gradients in 1-norm in high dimension.

**Proposition B.2.** *On Problem 2.1 with Zipf-distributed data (Assumption 2.3 with $\alpha = 1$), SD with simplified dynamics (Assumption 4.1) with $t_d(\tau) = \tau d^{1/2}/2$ and $\phi_d(\tau) = (1 + 1/\tau^2)^{-1}$ satisfies*

$$\frac{\mathcal{L}_d(\mathbf{W}_{t_d(\tau)}) - \mathcal{L}_d^*}{\mathcal{L}_d(\mathbf{W}_0) - \mathcal{L}_d^*} \overset{d}{\sim} \frac{1}{1 + \zeta(2\alpha)\tau^2}, \qquad \frac{\left\|\text{vec}(\nabla\mathcal{L}_d(\mathbf{W}_{t_d(\tau)}))\right\|_1}{\left\|\text{vec}(\nabla\mathcal{L}_d(\mathbf{W}_0))\right\|_1} \overset{d}{\sim} C\frac{d^{1/2}}{\log(d)\tau}\left(\frac{1}{\tau} + \frac{1}{\tau^3}\right)$$

*Proof.* Computations similar to Proposition 4.2 show that the 1-norm of the gradient is

$$\|\text{vec}(\nabla\mathcal{L}_d(\mathbf{W}_t))\|_1 = \sum_{k=1}^{k_*}(\pi_k - t\eta) + \sum_{k=k_*+1}^{d}\frac{\eta}{2}$$

where $k_*$ is the number of directions that are still in the decreasing regime after $T$ steps with stepsize $\eta$. As $\|\text{vec}(\nabla\mathcal{L}_d(\mathbf{W}_0))\|_1 = \sum_{k=1}^{d}\pi_k = 1$, this expression is also the normalized 1-norm of the gradient. Using the parameterization $\eta = 1/zt\phi$, where $z = \sum_{k=1}^{d}1/k$, we get the update

$$r_d(t) := \frac{\|\text{vec}(\nabla\mathcal{L}_d(\mathbf{W}_t))\|_1}{\|\text{vec}(\nabla\mathcal{L}_d(\mathbf{W}_0))\|_1} = \sum_{k=1}^{\lfloor\phi\rfloor}\left(\frac{1}{zk} - \frac{1}{z\phi}\right) + \sum_{k=\lfloor\phi\rfloor+1}^{d}\frac{1}{2tz\phi}$$

Using the same scaling as in Definition 4.4, $\phi_d(\tau) = (1 + 1/\tau^2)^{-1}$ and $2t_d(\tau) = \tau d^{1/2}$, we get

$$r_d(t_d(\tau)) \sim \frac{H_{1,\lfloor\phi_d(\tau)\rfloor}}{\log(d)} - \frac{\lfloor\phi_d(\tau)\rfloor}{\log(d)\phi_d(\tau)} + \frac{d - \lfloor\phi_d(\tau)\rfloor - 1}{\log(d)\phi_d(\tau)2t_d(\tau)} \sim \frac{d^{1/2}}{\log(d)}\left(\frac{1}{\tau} + \frac{1}{\tau^3}\right). \qquad \square$$

While getting a small error only requires scaling $t$ with $d^{1/2}$, getting the magnitude of the gradient in 1-norm smaller than a constant independent of $d$ requires scaling $t$ with $d/\log(d)$.

# C  Proofs for gradient descent

This section gives the proof of Theorem 3.1 for the scaling of gradient descent.

## C.1  Standard results

We start with standard results that are used in the subsequent proofs. The following classical relationships between sums and integrals of monotone functions will be used to bound the approximation error induced by analyzing the asymptotics of the integral instead of the sum.

**Lemma C.1** (Sum-Integral). *For a function $f$ that is monotone on $[a, b]$,*

$$\text{if } f \text{ is increasing on } [a,b], \qquad \sum_{i=a}^{b-1} f(k) \leq \int_a^b f(k)\, \mathrm{d}k \leq \sum_{i=a+1}^{b} f(k),$$

$$\text{if } f \text{ is decreasing on } [a,b], \qquad \sum_{i=a+1}^{b} f(k) \leq \int_a^b f(k)\, \mathrm{d}k \leq \sum_{i=a}^{b-1} f(k).$$

To apply these sum-integral relationships to the dynamics of gradient descent in Theorem 3.1, we need to describe when they are increasing or decreasing.

**Lemma C.2** (Unimodal sequence). *The sequence $s(k) = k^{-\alpha}(1 - k^{-\alpha})^t$ is non-negative on $k \geq 1$ and unimodal. It monotonically increases until $k_* = (1 + t)^{1/\alpha}$, then monotonically decreases.*

*Proof.* As $s(k)$ is non-negative, we can instead look at its logarithm,

$$\log s(k) = \log(N) - \alpha \log(k) + t \log(1 - k^{-\alpha}),$$

$$\frac{\partial}{\partial k} \log s(k) = \alpha t \frac{k^{-\alpha-1}}{1 - k^{-\alpha}} - \frac{\alpha}{k} = \frac{\alpha t}{k(k^\alpha - 1)} - \frac{\alpha}{k} = \frac{\alpha(t-1)(k^\alpha - 1)}{k(k^\alpha - 1)}.$$

The denominator is positive on $k \geq 1$, and the numerator is positive for small $k$ until the derivative changes sign at $\alpha t - \alpha(k^\alpha - 1) = 0$, or $k_* = (1 + t)^{1/\alpha}$. $\qquad\square$

At the partial sum $H_{d,\alpha} = \sum_{k=1}^d k^{-\alpha}$, appears in the proof of gradient and sign descent, we give its asymptotic behavior independently.

**Lemma C.3** (Normalizer Asymptotics). *As $d$ grows, the partial sum $H_{d,\alpha} = \sum_{k=1}^d k^{-\alpha}$ behaves as*

$$H_{d,\alpha} \overset{d}{\sim} \frac{1}{1-\alpha} d^{1-\alpha} \text{ if } \alpha < 1, \qquad H_{d,1} \overset{d}{\sim} \log(d) \qquad H_{d,\alpha} \overset{d}{\sim} \zeta(\alpha) \text{ if } \alpha > 1,$$

*where $\zeta$ is the zeta function, defined as the limit of $H_{d,\alpha}$, $\zeta(\alpha) = \sum_{k=1}^\infty k^{-\alpha} < \infty$ for $\alpha > 1$.*

*Proof.* For $\alpha > 1$, the sum converges to $\sum_{k=1}^\infty k^{-\alpha} = \zeta(\alpha)$. For $\alpha \leq 1$, the sum diverges as $d$ grows. As the sequence $k^{-\alpha}$ is decreasing in $k$, we can use the sum-integral bounds (C.1) to get

$$\int_1^{d+1} k^{-\alpha} \mathrm{d}k \leq \sum_{k=1}^d k^{-\alpha} \leq 1 + \int_1^d k^{-\alpha} \mathrm{d}k.$$

If $\alpha < 1$, the integrals evaluate to

$$\frac{\left((d+1)^{1-\alpha} - 1\right)}{1 - \alpha} \leq \sum_{k=1}^d k^{-\alpha} \leq \frac{\left(d^{1-\alpha} - 1\right) + 1}{1 - \alpha},$$

and both terms are asymptotically equivalent to $d^{1-\alpha}/(1 - \alpha)$ as $d \to \infty$. If $\alpha = 1$, this gives

$$\log(d + 1) \leq \sum_{k=1}^d k^{-\alpha} \leq \log(d) + 1.$$

Both terms are asymptotically equivalent to $\log(d)$. $\qquad\square$

The main purpose of the sum-integral bounds (C.1) and the Unimodal Lemma (C.2) is to bound on the error incurred by approximating the sum with the integral form of the loss.

**Lemma C.4** (Approximating error). *The approximation error between the following sum and integral,*

$$S_d(t) = \sum_{k=1}^{d} s(k) \qquad I_d(t) = \int_1^d s(k)\mathrm{d}k \qquad \text{where} \qquad s(k) = k^{-\alpha}(1 - k^{-\alpha})^{2t}$$

*can be bounded by the following error term,*

$$|S_d(t) - I_d(t)| \leq \delta_d(t) \quad \text{where} \quad \delta_d(t) := \begin{cases} \frac{1}{1+t}\left(1 - \frac{1}{1+2t}\right)^{2t} & \text{if } 1 + 2t \leq d^\alpha, \\ \frac{1}{d^\alpha}\left(1 - \frac{1}{d^\alpha}\right)^t & \text{if } 1 + 2t \geq d^\alpha. \end{cases} \tag{5}$$

*Proof.* By the Unimodal Lemma (C.2), the sequence $s(k)$ is increasing until $k_* = (1 + 2emt)^{1/\alpha}$ then decreasing, which lets us use the sum-integral bounds (C.1).

**For large $t$.** Suppose that $t$ is sufficiently large such that $k_* \geq d$ and $1 + 2t \geq d^\alpha$, meaning that the sequence $s(k)$ is increasing on $[1, d]$. Then,

$$\int_1^d s(k)\,\mathrm{d}k + s(1) \leq \sum_{1=1}^{d} s(k) \leq \int_1^d s(k)\,\mathrm{d}k + s(d). \tag{6}$$

Using that $s(1) = 0$ gives $|I_d(t) - S_d(t)| \leq s(d)$ when $t$ is large.

**For small $t$.** If $t$ is small and $k_* < d$ the sequence flips from increasing to decreasing on $[1, d]$. We still use the same idea, but bound the increasing and the decreasing subsequences separately.

**Upper bound.** As the sequences $s(k)$ in increasing on $[1, k_*]$ and decreasing on $[k_*, d]$,

$$\sum_{k=1}^{\lfloor k_* \rfloor - 1} s(k) \leq \int_1^{\lfloor k_* \rfloor} s(k)\,\mathrm{d}k, \qquad\qquad \sum_{k=\lfloor k_* \rfloor + 2}^{d} s(k) \leq \int_{\lfloor k_* \rfloor + 1}^d s(k)\,\mathrm{d}k.$$

Summing both bounds and adding the remaining terms $s(\lfloor k_* \rfloor), s(\lfloor k_* \rfloor + 1)$,

$$\sum_{k=1}^{d} s(k) \leq \int_1^{\lfloor k_* \rfloor} s(k)\,\mathrm{d}k + \int_{\lfloor k_* \rfloor + 1}^d s(k)\,\mathrm{d}k + s(\lfloor k_* \rfloor) + s(\lfloor k_* \rfloor + 1) \leq \int_1^d s(k)\,\mathrm{d}k + s(k_*),$$

where the last inequality uses the following simplifications,

$$\min\{s(\lfloor k_* \rfloor), s(\lfloor k_* \rfloor + 1)\} = \int_{\lfloor k_* \rfloor}^{\lfloor k_* \rfloor + 1} \min\{s(\lfloor k_* \rfloor), s(\lfloor k_* \rfloor + 1)\}\mathrm{d}k \leq \int_{\lfloor k_* \rfloor}^{\lfloor k_* \rfloor + 1} s(k)\mathrm{d}k,$$

$$\max\{s(\lfloor k_* \rfloor), s(\lfloor k_* \rfloor + 1)\} \leq s(k_*).$$

**Lower bound.** Now using the lower bound,

$$\int_1^{\lfloor k_* \rfloor} s(k)\,\mathrm{d}k \leq \sum_{k=2}^{\lfloor k_* \rfloor} s(k), \qquad\qquad \int_{\lfloor k_* \rfloor + 1}^d s(k)\,\mathrm{d}k \leq \sum_{k=\lfloor k_* \rfloor + 1}^{d-1} s(k).$$

Summing both bounds, we can complete the integral by adding and subtracting $\int_{\lfloor k_* \rfloor}^{\lfloor k_* \rfloor + 1} s(k)\,\mathrm{d}k$ and adding the remaining terms $s(1)$ and $s(d)$ to obtain

$$\sum_{k=1}^{\lfloor k_* \rfloor} s(k) \geq \int_1^d s(k)\,\mathrm{d}k - \int_{\lfloor k_* \rfloor}^{\lfloor k_* \rfloor + 1} s(k)\,\mathrm{d}k + s(1) + s(d) \geq \int_1^d s(k)\,\mathrm{d}k - s(k_*) + s(d),$$

where the last inequality uses that $s(1) = 0$, $s(k) \leq s(k_*)$.

**Combining the results** for the small $t$ regime gives

$$I_d(t) + s(k_*) \geq S_d(t) \geq I_d(t) - s(k_*) + s(d), \quad \text{so} \quad |I_d(t) - S_d(t)| \leq s(k_*).$$

**The final bound** in Eq. (5) expands $s(x) = x^{-\alpha}(1 - x^{-\alpha})^{2t}$ and replaces $k_*$ by $(1 + 2t)^{\frac{1}{\alpha}}$. $\qquad\square$

## C.2 Scaling laws for gradient descent

We are now ready to move to the proof of Theorem 3.1, for which we recall the theorem statement.

*Proof sketch.* We first give a sketch of the proof, which will be formalized in the next lemmas. Based on the reduced dynamics for gradient descent in Proposition A.1, we know that

$$r_d(t) = \frac{\mathcal{L}_d(t) - \mathcal{L}_d^*}{\mathcal{L}_d(0) - \mathcal{L}_d^*} = \frac{\sum_{k=1}^d k^{-\alpha}(1 - k^{-\alpha})^{2t}}{H_{d,\alpha}},$$

where $H_{d,\alpha} = \sum_{k=1}^d k^{-\alpha}$. Let $S_d$ and $I_d$ be the sum and integral variants of the denominator,

$$S_d(t) = \sum_{k=1}^d k^{-\alpha}(1 - k^{-\alpha})^{2t} \qquad\qquad I_d(t) = \int_1^d k^{-\alpha}(1 - k^{-\alpha})^{2t}\mathrm{d}k. \tag{7}$$

First, we establish in Lemma C.5 that the integral form converges to the rate $r(\tau)$ in Theorem 3.1,

$$\lim_{d\to\infty} \frac{I_d(t_d(\tau))}{H_{d,\alpha}} = r(\tau).$$

Next, we show in Lemma C.6 that the error incurred by approximating the sum $S_d$ by the integral $I_d$ is negligible, in the sense that $|I_d(t) - S_d(t)| \le \delta_d(t)$ and

$$\lim_{d\to\infty} \frac{\delta_d(t_d(\tau))}{I_d(t_d(\tau))} = 0 \quad \text{if } \alpha \le 1, \qquad \text{and} \qquad \lim_{\tau\to\infty}\lim_{d\to\infty} \frac{\delta_d(t)}{I_d(t)} = 0 \quad \text{if } \alpha > 1.$$

This gives the results that

$$r(\tau) = \lim_{d\to\infty} \frac{I_d(t_d(\tau))}{H_{d,\alpha}} \quad \text{if } \alpha \le 1, \qquad \text{and} \qquad r(t) \backsim \lim_{d\to\infty} \frac{I_d(t)}{H_{d,\alpha}} \quad \text{if } \alpha > 1.$$

with the values of $r(\tau)$ given in Theorem 3.1. $\qquad\qquad\qquad\qquad\qquad\qquad\qquad \square$

**Lemma C.5** (Asymptotics of the integrals). *Let $I_d(t)$ be the integral form given in Eq. (7) and $t_d(\tau)$ be the scaling given in Theorem 3.1. The following limits hold.*

If $\alpha < 1$,  $\quad t_d(\tau) = \frac{1}{2}\tau d^\alpha$,  $\quad \displaystyle\lim_{d\to\infty} \frac{I_d(t_d(\tau))}{H_{d,\alpha}} = \frac{1-\alpha}{\alpha} E_{\frac{1}{\alpha}}(\tau) \underset{\tau}{\sim} \frac{1-\alpha}{\alpha} \frac{e^{-\tau}}{\tau+1}$,

if $\alpha = 1$,  $\quad t_d(\tau) = \frac{1}{2}d^\tau$,  $\quad \displaystyle\lim_{d\to\infty} \frac{I_d(t_d(\tau))}{H_{d,\alpha}} = 1 - \tau \qquad$ *where* $\tau \in [0, 1]$,

if $\alpha > 1$,  $\quad t_d(\tau) = \tau$,  $\quad \displaystyle\lim_{d\to\infty} \frac{I_d(t_d(\tau))}{H_{d,\alpha}} = \frac{B\left(1 - \frac{1}{\alpha}, 1 + 2t\right)}{\alpha\zeta(\alpha)} \underset{\tau}{\sim} C\frac{1}{\tau^{1-\frac{1}{\alpha}}}\mathcal{L}_d(0)$,

*Proof.* **For $\alpha > 1$.** We use the change of variable $z = k^{-\alpha}$ to get

$$I_d(t) = \frac{1}{\alpha} \int_{d^{-\alpha}}^{1} z^{-\frac{1}{\alpha}}(1-z)^{2t}\mathrm{d}z$$

As $d \to \infty$, the integral converges to definition of the Beta function

$$\lim_{d\to\infty} \alpha I_d(t) = \int_0^1 z^{-\frac{1}{\alpha}}(1-z)^{2t}\,\mathrm{d}z =: B\left(1 - \frac{1}{\alpha}, 1 + 2t\right).$$

As $\lim_{d\to\alpha} H_{d,\alpha} = \zeta(\alpha) < \infty$ (Lemma C.3),

$$\lim_{d\to\infty} \frac{I_d(t)}{H_{d,\alpha}} = \frac{B\left(1 - \frac{1}{\alpha}, 1 + 2t\right)}{\alpha\zeta(\alpha)}.$$

As it is not easy to intuit the rate from the Beta function, we give an additional asymptotic equivalence for large $t$. Using Stirling's formula, the Beta function behaves as

$$B\left(1 - \frac{1}{\alpha}, 1 + 2t\right) \underset{t}{\sim} \Gamma\left(1 - \frac{1}{\alpha}\right)\frac{1}{(2t)^{1-\frac{1}{\alpha}}}.$$

**For $\alpha < 1$** we use the change of variable $z = 2tk^{-\alpha}$ to get

$$I_d(t) = \frac{1}{\alpha}(2t)^{\frac{1}{\alpha}-1} \int_{2td^{-\alpha}}^{2t} z^{-\frac{1}{\alpha}}\left(1 - \frac{z}{2t}\right)^{2t}\mathrm{d}z.$$

To have a well-defined integral, we need to introduce the scaling $2t_d(\tau) = \tau d^\alpha$,

$$I_d(\tau d^\alpha) = \frac{1}{\alpha}d^{1-\alpha}\tau^{\frac{1}{\alpha}-1} \int_{\tau}^{\tau d^\alpha} z^{-\frac{1}{\alpha}}\left(1 - \frac{z}{\tau d^\alpha}\right)^{\tau d^\alpha}\mathrm{d}z.$$

The factor of $d^{1-\alpha}$ will cancel out with the normalizer as $H_{d,\alpha} = \Theta(d^{1-\alpha})$ (Lemma C.3). The remaining integral should simplify for large $d$, as $(1 - z/\tau d^\alpha)^{\tau d^\alpha} \approx e^{-z}$, and converge to

$$\lim_{d\to\infty} \tau^{\frac{1}{\alpha}-1} \int_{\tau}^{\tau d^\alpha} z^{-\frac{1}{\alpha}}\left(1 - \frac{z}{\tau d^\alpha}\right)^{\tau d^\alpha}\mathrm{d}z = \tau^{\frac{1}{\alpha}-1}\int_{\tau}^{\infty} z^{-\frac{1}{\alpha}}e^{-z}\mathrm{d}z = E_{\frac{1}{\alpha}}(\tau),$$

where $E_p$ is the generalized exponential integral. To swap the limit and integral, we can verify that the dominated convergence theorem applies. The integral can be written as

$$\int_{\tau}^{\tau d^\alpha} z^{-\frac{1}{\alpha}}\left(1 - \frac{z}{\tau d^\alpha}\right)^{\tau d^\alpha} = \int_{\tau}^{\infty} a(z,d)\mathrm{d}z \quad \text{where} \quad a(z,d) := \mathbb{1}_{\{z \leq \tau d^\alpha\}} z^{-\frac{1}{\alpha}}\left(1 - \frac{z}{\tau d^\alpha}\right)^{\tau d^\alpha}.$$

The integrand $a(z,d)$ converges pointwise to $f(z) = z^{-\frac{1}{\alpha}}e^{-z}$ and is dominated by $f$ which is integrable as $\int_{\tau}^{\infty} f(z) = \tau^{1-\frac{1}{\alpha}} E_{\frac{1}{\alpha}}(\tau)$. Combined with the fact that $H_{d,\alpha} \overset{d}{\sim} d^{1-\alpha}/(1-\alpha)$, we get

$$\lim_{d\to\infty} \frac{I_d(\tau d^\alpha)}{H_{d,\alpha}} = \frac{1-\alpha}{\alpha} E_{\frac{1}{\alpha}}(\tau).$$

To simplify for large $\tau$ and obtain $E_{1/\alpha}(\tau) \underset{\tau}{\sim} e^{-\tau}/\tau$, we use the fact that the generalized exponential integral $E_p(z)$ in decreasing in $p$, meaning that $E_{\lfloor 1/\alpha \rfloor}(\tau) > E_{1/\alpha}(\tau) > E_{\lceil 1/\alpha \rceil}(\tau)$, and that for

integer values of $p$ we have $e^{-\tau}/\tau+n \le E_n(\tau) \le e^{-\tau}/\tau+n-1$ ([DLMF, §8.19(ix)](#)). Both bounds are asymptotically equivalent to $e^{-\tau}/(\tau+1)$.

**For $\alpha = 1$** we use the change of variable $k = d^z$ or $z = \log_d(k)$ to get

$$I_d(t) = \log(d) \int_0^1 \left(1 - d^{-z}\right)^{2t} \mathrm{d}z.$$

The normalizer scales as $H_{d,\alpha} \overset{d}{\sim} \log(d)$ ([Lemma C.3](#)) so only the integral remains. To make meaningful progress, we introduce the scaling $2t_d(\tau) = d^\tau$ for $\tau \in [0,1]$,

$$\frac{I_d(d^\tau)}{\log(d)} = \int_0^1 \left(1 - \frac{d^{\tau-z}}{d^\tau}\right)^{d^\tau} \mathrm{d}z.$$

As $d \to \infty$, the integrand converges to $0$ if $z \in (0, s)$ and to $1$ if $z \in (s, 1)$, and is dominated by $f(x) = 1$ so by the DCT we can swap the limit and integral to get

$$\lim_{d\to\infty} \frac{I_d(d^\tau)}{H_{d,\alpha}} = \lim_{d\to\infty} \int_0^1 \left(1 - \frac{d^{\tau-z}}{d^\tau}\right)^{d^\tau} \mathrm{d}z = \int_0^\tau 0 \mathrm{d}z + \int_\tau^1 1 \mathrm{d}z = 1 - \tau. \qquad \square$$

**Lemma C.6** (Approximation error is negligible). *Let $\delta_d(t)$ be the upper bound on the approximation error derived in the [Approximation Error Lemma (C.4)](#). We have that*

$$\lim_{d\to\infty} \frac{\delta_d(t_d(\tau))}{I_d(t_d(\tau))} = 0 \text{ if } \alpha \le 1, \qquad and \qquad \lim_{\tau\to\infty} \lim_{d\to\infty} \frac{\delta_d(t)}{I_d(t)} = 0 \text{ if } \alpha > 1.$$

*Proof.* Recall that the bound approximation error $\delta$ in [Approximation Error Lemma (C.4)](#) is

$$|S_d(t) - I_d(t)| \le \delta_d(t) \quad \text{where} \quad \delta_d(t) := \begin{cases} \frac{1}{1+2t}\left(1 - \frac{1}{1+2t}\right)^{2t} & \text{if } 1 + 2t \le d^\alpha, \\ \frac{1}{d^\alpha}\left(1 - \frac{1}{d^\alpha}\right)^t & \text{if } 1 + 2t \ge d^\alpha. \end{cases}$$

**For $\alpha > 1$,** $t$ does not scale with $d$ so we are in the small $t$ regime, $1 + 2t \le d^\alpha$. In this regime,

$$\delta_d(t) = \frac{1}{2t+1}\left(1 - \frac{1}{2t+1}\right)^{2t} \le \frac{1}{2t+1}.$$

The error $\delta_d(t)$ does not vanish with $d$, but it goes down as $O(1/t)$. As the integral $I_d(t)$ is of order $\Theta(1/t^{1-\frac{1}{\alpha}})$, the relative error is of order $O(1/t^{\frac{1}{\alpha}})$, and vanishes for large $t$.

**For $\alpha < 1$,** we scale $t$ with $d$ as $2t = \tau d^\alpha$. Whether $t$ is small or large depends on $\tau$. If $\tau < 1$, we are in the small $t$ regime as $1 + \tau d^\alpha \le d^\alpha$ and

$$\delta_d(\tau d^\alpha) = \frac{1}{\tau d^\alpha + 1}\left(1 - \frac{1}{\tau d^\alpha + 1}\right)^{\tau d^\alpha} \le \frac{1}{\tau d^\alpha}.$$

If $\tau \ge 1$ we are in the large $t$ regime and

$$\delta_d(\tau d^\alpha) = \frac{1}{d^\alpha}\left(1 - \frac{1}{d^\alpha}\right)^{\tau d^\alpha} \le \frac{1}{d^\alpha}.$$

In both cases $\lim_{d\to\infty} \delta_d(\tau d^\alpha) \to 0$ and the relative error also vanishes.

**For $\alpha = 1$** we scale $t$ with $d$ as $2t = d^\tau$ for $\tau \in [0,1]$. Taking $d \to \infty$ puts us in the small $t$ regime, $1 + 2t = 1 + d^\tau \le d$. In this regime,

$$\delta_d(d^\tau) = \frac{1}{d^\tau + 1}\left(1 - \frac{1}{d^\tau + 1}\right)^{d^\tau} \le \frac{1}{d^\tau},$$

which also vanishes with $d$. $\qquad \square$

# D  Proofs for sign descent

This section gives the derivation for the scaling of time and the step-size for sign descent given in Definition 4.4 and the resulting asymptotic convergence rates of Theorem 4.5. Each result start from the relative loss defined as follows.

**Definition D.1** (Normalized loss for sign descent). *Let $\mathcal{L}_d(t, \eta)$ be the loss after with step-size $\eta$ as defined in Proposition 4.2, and $\eta(T, \phi) = 1/H_{d,\alpha}T\phi^\alpha$ be the reparameterization of the step-size derived from Proposition 4.3. The relative loss after $T$ steps of the simplified sign descent dynamics on Problem 2.1 with power-law frequencies as in Assumption 2.3 is*

$$r_d(T, \phi) := \frac{\mathcal{L}_d(T, \eta(T, \phi)) - \mathcal{L}_d^*}{\mathcal{L}_d(0) - \mathcal{L}_d^*} = \frac{H_{\lfloor\phi\rfloor,2\alpha} - 2H_{\lfloor\phi\rfloor,\alpha}\phi^{-\alpha} + \lfloor\phi\rfloor\phi^{-2\alpha} + \frac{d-\lfloor\phi\rfloor}{4T^2}\phi^{-2\alpha}}{H_{d,2\alpha}},$$

*where $H_{n,p} = \sum_{k=1}^n k^{-p}$.*

*Proof.* Starting from Proposition 4.2 and using the fact that, if $\phi \in [1, d]$, the number of components in the decreasing phase of the simplified sign descent dynamics is $\lfloor\phi\rfloor$, we expand the square and replacing the sums by $H_{n,p}$,

$$r_d(T, \phi) = \frac{\sum_{k=1}^{\lfloor\phi\rfloor}(k^{-\alpha} - \phi^{-\alpha})^2 + \sum_{k=\lfloor\phi\rfloor+1}^d \left(\frac{1}{2T\phi^\alpha}\right)^2}{\sum_{k=1}^d k^{-2\alpha}},$$

$$= \frac{\left(\sum_{k=1}^{\lfloor\phi\rfloor} k^{-2\alpha} - 2k^{-\alpha}\phi^{-\alpha} + \phi^{-2\alpha}\right) + \frac{d-\lfloor\phi\rfloor}{4T^2}\phi^{-2\alpha}}{\sum_{k=1}^d k^{-2\alpha}},$$

$$= \frac{H_{\lfloor\phi\rfloor,2\alpha} - 2H_{\lfloor\phi\rfloor,\alpha}\phi^{-\alpha} + \lfloor\phi\rfloor\phi^{-2\alpha} + \frac{d-\lfloor\phi\rfloor}{4T^2}\phi^{-2\alpha}}{H_{d,2\alpha}}. \qquad \square$$

Our rates are given for a choice of scaling of the step-size $\phi_d(\tau)$ and time $T_d(\tau)$, as

$$r(\tau) := \lim_{d\to\infty} r_d(T_d(\tau), \phi_d(\tau)).$$

## D.1  Scaling of sign descent for $\alpha = 1/2$

**Proposition D.2.** *For the relative loss defined in Definition D.1, if $\alpha = 1/2$, the scalings*

$$T_d(\tau) = \tfrac{1}{2}d^{\frac{1}{2}\tau}, \qquad\qquad \phi_d(\tau) = d^{1-\tau},$$

*are obtained by setting $\phi_d(\tau) = d^{x_*(\tau)}$ where $x_*(\tau)$ is the solution to*

$$x_*(\tau) = \arg\min_{0<x\leq 1} \lim_{d\to\infty} r_d(T_d(\tau), d^x).$$

*These choices result in the scaling $r(\tau) = 1 - \tau$.*

*Proof.* We start from the normalized loss given $\phi$,

$$r_d(T, \phi) = \frac{H_{\lfloor\phi\rfloor,1} - 2H_{\lfloor\phi\rfloor,\frac{1}{2}}\phi^{-\frac{1}{2}} + \lfloor\phi\rfloor\phi^{-1} + \frac{d}{4T^2}\phi^{-1} - \frac{1}{4T^2}\lfloor\phi\rfloor\phi^{-1}}{H_{d,1}}.$$

Taking $4T^2 = d^\tau$ and $\phi = d^{1-\tau}$, most terms vanish as $d \to \infty$ as $H_{n,\frac{1}{2}} \sim 2\sqrt{n}$, $H_{n,1} \sim \log(n)$, and

$$\frac{2H_{\lfloor d^{1-\tau}\rfloor,\frac{1}{2}}d^{-\frac{1-\tau}{2}}}{H_{d,1}}, \frac{\lfloor d^{1-\tau}\rfloor d^{-(1-\tau)}}{H_{d,1}}, \frac{1}{H_{d,1}}, \frac{\lfloor d^{1-\tau}\rfloor d^{-1}}{H_{d,1}} \text{ are all } \Theta\left(\frac{1}{\log(d)}\right) \text{ and converge to } 0.$$

The first term is the only one remaining, and gives the scaling

$$\lim_{d\to\infty} r_d(T(d, \tau), d^x) = \lim_{d\to\infty} \frac{H_{\lfloor d^{1-\tau}\rfloor,1}}{H_{d,1}} = \lim_{d\to\infty} \begin{cases} x & \text{if } 1 - \tau \leq x, \\ \infty & \text{otherwise.} \end{cases}$$

The optimum is at $x_*(\tau) = 1 - \tau$ and gives $r(\tau) = \lim_{d\to\infty} r_d(T(d, \tau), d^{1-\tau}) = 1 - \tau$. $\qquad \square$

## D.2 Scaling of sign descent for $\alpha < 1/2$

**Proposition D.3.** *For the relative loss defined in [Definition D.1](#), if $\alpha < 1/2$, the scalings*

$$T_d(\tau) = \tau, \qquad \phi_d(\tau) = \begin{cases} d & \text{if } \tau \leq \sqrt{\frac{1-c_1}{4c_2}}, \\ d\big(c_1 + c_2 4\tau^2\big)^{-1} & \text{otherwise}, \end{cases}$$

*where $c_1 = 1 - \frac{1}{2\alpha}$ and $c_2 = \frac{\alpha}{\alpha-1}$, are obtained by setting $\phi_d(\tau) = dx_*(\tau)$ where*

$$x_*(\tau) = \arg\min_{0<x\leq 1} \lim_{d\to\infty} r_d(T_d(\tau), dx).$$

*These choices result in the scaling*

$$r(\tau) = \begin{cases} 2\alpha c_2 & \text{if } \tau \leq \sqrt{\frac{1-c_1}{4c_2}} \\ \frac{(c_1+c_2 4\tau^2)^{2\alpha}}{4\tau^2} & \text{otherwise} \end{cases} \;\sim\; c_2^{2\alpha} \frac{1}{(2\tau)^{2-4\alpha}}.$$

*Proof.* Substituting $\phi = dx$, taking the limit as $d \to \infty$, and using that $H_{d,p} \sim \frac{d^{1-p}}{1-p}$ for $p < 1$, define $f_\tau(x)$ as the limit of $r_d(\tau, dx)$ as $d$ grows,

$$f_\tau(x) = \lim_{d\to\infty} r_d(\tau, dx) = \lim_{d\to\infty} \frac{H_{\lfloor dx\rfloor,2\alpha} - 2H_{\lfloor dx\rfloor,\alpha}(dx)^{-\alpha} + \lfloor dx\rfloor(dx)^{-2\alpha} + \frac{d-\lfloor dx\rfloor}{4\tau^2}(dx)^{-2\alpha}}{H_{d,2\alpha}},$$

$$= \frac{\frac{1}{1-2\alpha}x^{1-2\alpha} - 2\frac{1}{1-\alpha}x^{1-2\alpha} + x^{1-2\alpha} + \frac{1}{4\tau^2}x^{-2\alpha} - \frac{1}{4\tau^2}x^{1-2\alpha}}{\frac{1}{1-2\alpha}}.$$

We will show that our choice of step-size corresponds to taking $r(\tau) = \min_{0<x\leq 1} f_\tau(x)$. Gathering terms, $f_\tau(x)$ is proportional to the following polynomial

$$f_\tau(x) \propto x^{1-2\alpha}\left(1 + \frac{1}{1-2\alpha} - 2\frac{1}{1-\alpha} - \frac{1}{4\tau^2}\right) + \frac{1}{4\tau^2}x^{-2\alpha},$$

which has a unique stationary point at

$$x_{\text{stat}}(\tau) = \frac{2\alpha}{4\tau^2}\frac{1}{(1-2\alpha)\left(1 + \frac{1}{1-2\alpha} - 2\frac{1}{1-\alpha} - \frac{1}{4\tau^2}\right)} = \left(1 - \frac{1}{2\alpha} + \frac{\alpha}{1-\alpha}4\tau^2\right)^{-1}.$$

If $x_{\text{stat}}(\tau) \notin (0,1]$, we know the function $f$ is decreasing on $[0,1]$ as $\lim_{x\to 0} f_\tau(x) = \infty$, $f_\tau(1)$ is finite, and there is no stationary point in $(0,1])$, $r(\tau) = f_\tau(1)$. If the stationary point is in $(0,1]$, it is the minimum as $f_\tau$ must be decreasing from 0 to $x_{\text{stat}}(\tau)$. This gives

$$x_* = \arg\min_{0<x\leq 1} f_\tau(x) = \begin{cases} x_{\text{stat}}(\tau) & \text{if } 0 < x_{\text{stat}}(\tau) \leq 1, \\ 1 & \text{otherwise.} \end{cases}$$

and $0 < x_{\text{stat}}(\tau) \leq 1$ is equivalent to $\tau \geq \frac{1}{2}\sqrt{\frac{1-\alpha}{2\alpha^2}}$. If $\tau \geq \frac{1}{2}\sqrt{\frac{1-\alpha}{2\alpha^2}}$ and $x_*(\tau) = 1$, we get

$$f_\tau(x_*(\tau)) = 1 - 2\frac{1-2\alpha}{1-\alpha} + (1-2\alpha) = 2\frac{\alpha^2}{1-\alpha}.$$

If $\tau < \frac{1}{2}\sqrt{\frac{1-\alpha}{2\alpha^2}}$ and $x_*(\tau) = \left(1 - \frac{1}{2\alpha} + \frac{\alpha}{1-\alpha}4\tau^2\right)^{-1}$ we get

$$f_\tau(x_*(\tau)) = (1-2\alpha)\left(x^{1-2\alpha}\left(1 + \frac{1}{1-2\alpha} - 2\frac{1}{1-\alpha} - \frac{1}{4\tau^2}\right) + \frac{1}{4\tau^2}x^{-2\alpha}\right),$$

$$= \frac{\left(1 - \frac{1}{2\alpha} + \frac{\alpha}{1-\alpha}4\tau^2\right)^{2\alpha}}{4\tau^2},$$

which can be simplified for large $\tau$ as $f_\tau(x_*(\tau)) \sim \left(\frac{\alpha}{1-\alpha}\right)^{2\alpha}\frac{1}{(2\tau)^{2-4\alpha}}$. $\qquad\square$

### D.3 Scaling of sign descent for $\alpha > 1/2$

For $\alpha > 1/2$, the expression for the loss does not simply as $d \to \infty$. The conditional frequencies decay fast, meaning that most of the loss comes from the few high-frequency words. As a result, we cannot define the scaling of the step-size as the minimization problem for the optimal scaling in the limit $d \to \infty$. Instead, we use the fact that the (normalized) loss can not converge to 0 unless all components enter the oscillatory regime, at which point we can compute an optimal step-size.

**Proposition D.4.** *For the relative loss defined in [Definition D.1](#), if $\alpha > 1/2$ and $4T^2 \geq \frac{d-1}{2^\alpha - 1}$, the optimal-step size is given by*

$$\phi_*(d, T) = \arg\min_\phi r_d(T, \phi) = \left(1 + \frac{d-1}{4T^2}\right)^{1/\alpha}.$$

*This gives the following scaling for $\tau^2 > 1/(2^\alpha - 1)$*

$$T_d(\tau) = \tau \tfrac{1}{2}\sqrt{d}, \qquad \phi(\tau) = \left(1 + \frac{1}{\tau^2}\right)^{1/\alpha}, \qquad r(\tau) = \frac{1}{\zeta(2\alpha)} \frac{1}{1 + \tau^2}.$$

*Proof.* If $\phi \geq 2$, the normalized loss is lower-bounded by the error on the first two components,

$$r_d(T, \eta(T, \phi)) = \frac{\sum_{k=1}^{\lfloor \phi \rfloor} (k^{-\alpha} - \phi^{-\alpha})^2 + \sum_{k=\lfloor \phi \rfloor + 1}^{d} \left(\frac{1}{2T\phi^\alpha}\right)^2}{H_{d, 2\alpha}}.$$

This is lower-bounded by a constant $C > 0$ independently of $T$, and implies that we cannot make progress by running longer unless $\phi < 2$. If only the first component is oscillating, the optimal $\phi$ is

$$\phi_*(d, T) = \arg\min_\phi r_d(T, \eta(T, \phi)) = \arg\min_\phi (1 - \phi^{-\alpha})^2 + \frac{d-1}{4T^2} \phi^{-2\alpha} = \left(1 + \frac{d-1}{4T^2}\right)^{1/\alpha}.$$

To be consistent with only having two components oscillating, this requires $\phi_*(d, T) \leq 2$, giving the constraint that this only holds when $(1 + \frac{d-1}{4T^2})^{1/\alpha} \leq 2$ or $4T^2 \geq \frac{d-1}{2^\alpha - 1}$. Taking the scaling $4T_d(\tau)^2 = \tau^2 d$ gives the limit

$$\phi(\tau) = \lim_{d \to \infty} \phi_*(d, T_d(\tau)) = \left(1 + \frac{1}{\tau^2}\right)^{1/\alpha} \quad \text{if} \quad \tau^2 > \frac{1}{2^\alpha - 1},$$

and the asymptotic loss

$$\lim_{d \to \infty} r_d(T_d(\tau, d), \phi(\tau)) = \frac{(1 - \phi(\tau)^{-\alpha})^2 + \frac{1}{\tau^2} \phi(\tau)^{-2\alpha}}{\zeta(2\alpha)},$$

$$= \frac{\left(1 - \left(1 + \frac{1}{\tau^2}\right)^{-1}\right)^2 + \frac{1}{\tau^2}\left(1 + \frac{1}{\tau^2}\right)^{-2}}{\zeta(2\alpha)} = \frac{1}{1 + \tau^2} \frac{1}{\zeta(2\alpha)},$$

where $H_{d, 2\alpha} \xrightarrow{d} \zeta(2\alpha)$, the Riemann zeta function. $\qquad\square$

[Proposition D.4](#) and [Theorem 4.5](#) only gives guarantees for the regime $\tau^2 > 1/(2^\alpha - 1)$. The extension of the scalings to the regime $\tau^2 \leq 1/(2^\alpha - 1)$ was decided arbitrarily to fit empirical data. To fit the empirical the empirical data when both $\tau$ and $\alpha$ are small ($\alpha \leq 1$), the asymptotic scaling presented in [Theorem 4.5](#) uses the following step-size scaling

$$\tilde{\phi}(\tau) = \begin{cases} \left(1 + \frac{1}{\tau^2}\right) & \text{if } \tau^2 < (2^\alpha - 1)^{-1} \text{ and } \alpha < 1, \\ \left(1 + \frac{1}{\tau^2}\right)^{1/\alpha} & \text{otherwise}, \end{cases} \quad \text{instead of} \quad = \left(1 + \frac{1}{\tau^2}\right).$$

and the following approximation for the loss,

$$r_d(T_d(\tau, d), \phi(\tau)) \overset{\tau, d}{\sim} \frac{1}{1 + \zeta(2\alpha)\tau^2} \quad \text{instead of} \quad \frac{1}{1 + \tau^2} \frac{1}{\zeta(2\alpha)}.$$

Both expressions are asymptotically equivalent as $d \to \infty$ and $\tau \to \infty$, but the above proposals (given in [Definition 4.4](#)) fit the observed best step-size and loss scalings better.

