# OpenReview forum: "Scaling Laws for Gradient Descent and Sign Descent for Linear Bigram Models under Zipf’s Law"
_NeurIPS.cc/2025/Conference — NeurIPS 2025 poster_

### Official Review · Reviewer_G8T2 · 2025-06-02

**Clarity:** 3
**Significance:** 3
**Originality:** 4
**Rating:** 5
**Confidence:** 4

**Summary:**

The paper investigates scaling laws for the convergence of Gradient Descent (GD) and Sign Descent (SD) optimizers for a simplified linear bigram model trained on data that follows power law distributions 1/k^α (in both token frequency, conditional token frequency, and consequently eigenvalues of the Hessian), where k is the frequency rank of each token.  This simplified model is chosen because it allows closed-form solutions for both GD and SD, while still reproducing differences between GD and SD seen in practice.  The final deliverable is a set of empirically-validated scaling theorems that show how many iterations are required to reach a relative loss target ϵ as a function of (1) the dimensionality and (2) the exponent of the power law α. GD is shown to perform particularly poorly on Zipfian α=1 data (for small ϵ), scaling almost linearly in the dimensionality, whereas SD scales closer to the square root of the dimensionality.  These findings formalize some prior empirical observations regarding the impact of heavy-tailedness of the data (as Zipfian reflects real language data).

**Questions:**

Could you make the connection to LLM studies a bit more clear?  I.e., do modern methods of creating LLM vocabularies avoid the degenerate case where a specific token is always followed by a specific other token (e.g., "et" and "cetera" being two separate tokens), thus perhaps helping justify your power law assumptions?  Recent work has also shown that increasing vocab size with scale can improve loss, which effectively means that changing the vocab size can change the slope of the scaling law.  I wonder how this relates to your work.

Motivating α < 1: Can you provide stronger justification or evidence from practical datasets (beyond Figure 3) that data following α < 1 is worth studying, and thus your results may be practically meaningful?  Is this something that arises outside of language?  In other words, why do we care about very, very heavy tails?

Hyperparameter implications: You mention your scaling laws can guide hyperparameter selection as vocabulary size increases - can you elaborate precisely on how practitioners might use these scaling laws to adjust specific hyperparameters (e.g., step size or optimizer choice)?

Generalizability: How do you anticipate your results might generalize beyond linear bigram models (e.g., to transformers or non-squared-loss settings)? Which of your assumptions are most critical and least likely to hold in more complex models?  Do you have further thoughts on the first/last layer issues seen previously?  Do your findings imply things about parameters in non first/last Transformer layers?  Does the theory in this work make any testable predictions for LLMs?

Conditional distribution deviations: Can you quantify or bound how deviations from your conditional power-law assumption might impact your theoretical predictions?

**Ethical Concerns:**

["NO or VERY MINOR ethics concerns only"]

**Final Justification:**

I have read the other reviews and the rebuttal from the authors.  I find this is a technically solid paper, which, despite its assumptions, helps advance our understanding regarding optimization of natural language.  I believe the authors can and will take all the detailed reviews and feedback and use them to further improve the final version of the paper.  This is why I am recommeding "Accept"

**Limitations:**

Yes

**Quality:**

3

**Strengths And Weaknesses:**

Strengths:

Quality:
- Technical soundness: The theoretical derivations are rigorous and are supported by experimental results on both synthetic (controlled α) and real data (OpenWebText).
- Supported claims: The results on GD and SD scaling are clearly demonstrated experimentally (Figure 2). Their simplified theoretical model aligns well with empirical data.

Clarity:
- Structure: Generally well-organized; contributions clearly stated.
- Figures: Good illustrations effectively show theoretical predictions against empirical outcomes.

Significance:
- Key insight: Shows explicitly why and when Sign Descent (proxy for Adam-like methods) outperforms Gradient Descent for power-law-distributed data.
- Practical Impact: The analysis provides insights into optimizer performance with increasing vocabulary size, which is practically relevant for training language models.  However, because of the many assumptions and restrictions needed for the study, I do not foresee an immediate impact on practical LLM training arising directly from this work.

Originality:
- Novelty of results: The paper uniquely addresses a gap by studying heavier-tailed data (α ≤ 1), which previous literature largely ignores.

Weaknesses:

Scope limitation:
- As noted above, the analysis is restricted to the simple linear bigram model with squared loss.  A number of other assumptions are made. While informative, the impact may not directly generalize to complex transformer layers or cross-entropy objectives common in practical NLP applications.
- The assumption that conditional distributions also follow the same power-law as the marginal frequencies (Assumption 2.3) seems idealized. The authors investigate and acknowledge deviations (Figure 3), but the implications for practical scenarios are less clear. It's also unclear if α < 1 cases are practically motivated, as most natural language settings specifically reflect α ≈ 1.  This is not really investigated or discussed.

Clarity:

The paper would benefit greatly (and potentially have larger impact) by adding more general topic sentences and definitions at the beginning of sections. These can serve to anchor and provide scaffolding for readers who are not experts in this area (but may generally be interested in scaling laws, optimization, etc.).  That is, this paper may serve as a good entry point for such readers and therefore I would recommend greater clarity and also perhaps an extended related work section in an appendix to clarify the progress in understanding that has been made in this area over the last several years.

As an example of something requiring clarification, I note that the term “scaling law” is mentioned in the title, abstract, intro, Figure 1, related work, and if we jump ahead to the conclusion.  But none of this actually defines what an optimization scaling law actually is?  Readers familiar with scaling laws in other contexts will expect a clear and precise definition: here we consider a law that looks at how X changes in Y and Z (or whatever).
- The related work mentions scaling laws for parameters like N and D (in C, for compute-efficient training) and η with B (for well-tuned training), but on a first pass, I wasn't able to extract similar scaling laws for optimization.
- These sections of the paper definitely hint at what form the scaling law might take, but I think the paper’s clarity would have really benefited if the introduction had made the specific scaling law clear, which seems to happen at the start of Section 2.2: “Our goal is to derive scaling laws for the loss of Problem 2.1 in d dimensions after t steps, Ld(t), as d → ∞.”
- So, in the intro, you could say something like, “specifically, we look at how the [squared-error] loss of a [linear] model, of dimensionality d, scales [after t steps] for tokens generated by a bigram LM with power law exponent α.” (where the bracketed parts depends on how specific you want to be; the key is to mention L scaling in d). You could also work this into the caption of Figure 1 in some form.

Also, it would have helped on a first read of the paper if we had more clearly defined what is meant by eigenvalues of the “data” (i.e., eigenvalues of the Hessian), and how the power law in these eigenvalues specifically relate to the power law in the unigram (and bigram) frequencies.  While you use α for both in the intro, the full connection wasn’t clear to me until after I read Section 2.  I definitely assumed that we were talking about the same α, but you could have helped by saying something like, “in our case, the eigenvalues follow the same power law as the bigram frequencies, thus we naturally consider cases where α < 1 by virtue of ...” or whatever.

Other clarity issues:
- 1.2 “Given matrices X and Y ∈ {0, 1}^{n×d} containing the one-hot encodings of n pairs of tokens”.  This was actually hard to parse at first (reading it as X and Y *each* contain pairs of tokens?).  Can we slow down a little bit here, say that x is a 1-hot-encoding of a token, and y is a 1-hot-encoding of the NEXT token, and we have a dataset of n such pairs (next-token prediction), which we put into matrix form as X and Y.  That would have helped a lot.
- 2.1: Can we be a bit more explicit rather than defining things as we go?  Like, “let x* be the solution (loss minimizer?) of a convex quadratic loss in reduced form, f(x).  Let … be the eigenvectors/eigenvalues of A.  The loss …”
- Figure 3, right: this plot was a bit confusing on first read.  The following words help: “once sorted, the next-word frequencies also follow a power law with the same exponent,” but the xlabel says “Rank of word k”, not “Rank of word k among followers of word j,” which I think is really meant.

Very minor for the final paper:
- Note it does say to “delete this instruction block” in the NeurIPS checklist
- “Our approach is inspired by the line of work on theoretical scaling laws, or asymptotic convergence as the dimensionality grows” – could you revise to clarify the semantics of “or” here, i.e., does this mean “also known as” or are you talking about two distinct areas of work?

---

> ### Author Rebuttal · Authors · 2025-07-30
>
> # General response
>
> We thank you for your efforts in this review and for the constructive feedback that will improve the paper.
>
> Multiple reviewers raised similar points regarding the simplicity of the model. We start by addressing those points, and then answer the questions specific to your review. We invite you to read the response to the other reviews for more information on the following points.
> - [Response to Reviewer pUyz](https://openreview.net/forum?noteId=uPDWwfJLys): **Cross-entropy loss vs. squared loss**
>   Capturing the dynamics of optimization on the cross-entropy loss is not possible, but experiments show that the performance gap between gradient descent (GD) and sign descent (SD) as $d$ increases is similar.
> - [Response to Reviewer xbaC](https://openreview.net/forum?noteId=pIZjJafrGp): **Averaging oscillations with Assumption 4.1 and Adam vs. sign descent**
>   SD is a simple proxy for Adam, but does capture the main benefit of a better scaling with $d$. Experimentally, Adam performs better than SD and also scales better than GD in $d$.
> - [Response to Reviewer 8S6d](https://openreview.net/forum?noteId=Ezuh3XcjFB): **Relation to prior scaling laws**
>   The main novelty of our approach is on the scaling of $t$ with $d$ to derive a scaling law for GD when the difficulty of the problem scales with $d$.
> - [Response to Reviewer 9d42](https://openreview.net/forum?noteId=3xCuAZLexY): **Axis-aligned model and sign descent**
>   The axis-aligned property of the model is a key assumption that allows not only for the analysis of SD but is also crucial to get an improvement over GD, and this property mimics the axis-alignment found in the first and last layers of LLMs.
> - [Response to Reviewer G8T2](https://openreview.net/forum?noteId=CfxXnAH6I3): **Connections to LLM training**
>   Discussion of the bigram frequencies for real tokenizers and possible insights on LLM training.
>
> We ran additional experiments to answer those questions, but as the NeurIPS guidelines do not allow sharing figures or updating the paper, we can only share a written description. We will use the additional page given for camera ready to include those experimental results and incorporate the clarifications and feedback provided by the reviewers.
>
> ## Simplicity of the model
>
> > Training language models uses softmax cross‑entropy, not squared error. [...]
> > _Reviewer pUyz_
> > Some of the assumptions are fairly unrealistic. Eg, the model is linear, uses square loss, assumes deterministic full-batch updates, and collapses Adam to SD [...]
> > _Reviewer xbaC_
> > the analysis is restricted to the simple linear bigram model with squared loss [which] may not directly generalize to complex transformer layers or cross-entropy objectives [...]
> > _Reviewer G8T2_
>
> Multiple reviewers have noted that the model we study is simple, and perhaps too simple to be of practical use for the training of LLMs.
> We agree with this assessment.
> We do not claim that our theory is directly applicable to LLMs.
> Understanding the impact of other hyperparameters is an interesting next step.
> The choice of loss function, mini-batch size, momentum, model (attention or normalization layers), or the study of generalization instead of training error are all likely to yield new insights.
> However, we believe the simplicity of the model is a strength of the paper.
>
> Despite its simplicity, the model is not trivial and already captures a complex behavior observed in practice, that SD outperforms GD for language data.
> Its simplicity allows us to establish clean theoretical results for the convergence of GD and SD without relying on random matrix theory, Fourier transforms, or other complex mathematical tools. The scaling laws provide an explanation for the improvement of SD over GD and ties it to the properties of the data through the heavy tails in the distribution of the eigenvalues. These results prove that the performance gap grows with the vocabulary size even on a linear regression model.
>
> We have ran additional experiments to show that the conclusion that SD scale better than GD extend to the cross-entropy loss and the use of Adam instead of SD. For space reasons, we cannot include these results in every response. We invite you to read [the response to Reviewer pUyz](https://openreview.net/forum?noteId=uPDWwfJLys) for more details on the cross-entropy loss, and [the response to Reviewer xbaC](https://openreview.net/forum?noteId=pIZjJafrGp) for more details on Adam.
>
> ---
>
> # Points raised in your review
>
> We now address the specific questions raised in your review.
>
> ## Bigram distribution and LLMs
> > The assumption that conditional distributions also follow the same power-law as the marginal frequencies (Assumption 2.3) seems idealized.
> > Do [tokenizers] avoid the degenerate case where a specific token is always followed by a specific other token [e.g., "et" and "cetera",] helping justify your power law assumptions?
>
> We agree that our model is an approximation and not an accurate model of language. But the marginal and conditional probabilities do approximately follow the power-law, as shown using a standard tokenizer on OpenWebText in Figure 3. Most next-word distributions are far from being supported on just a few tokens.
>
> Anecdotally, from our trained vocabulary, the next-word distributions that deviate most from Zipf's law appear to be numbers (most likely followed by other numbers) and special characters used to start the BPE tokenizer (very rare compared to other tokens, some not appearing in the data). While the support of the next-word distribution is not the entire vocabulary, the tokens generated by the algorithm are more in line with the power-law assumption and can be followed by $10^3$ different tokens or more.
>
> > Can you quantify or bound how deviations from your conditional power-law assumption might impact your theoretical predictions?
>
> This would require assumptions on the form of the deviations. Following Velikanov and Yarotsky (2022), assuming that the frequencies are bounded by $1/k^\alpha$ up to a constant seems applicable given Figure 3 and might be sufficient to derive upper bounds on the loss instead of an asymptotic equivalence.
>
> ## Connection to existing work on vocabulary size
> > Recent work has also shown that increasing vocab size with scale can improve loss, which effectively means that changing the vocab size can change the slope of the scaling law
>
> Existing works shows experimentally that larger vocabulary sizes are beneficial when the FLOPS budget is large (Tao et al. 2024). Whether this is because they need more FLOPS per iteration or more iterations is unclear. Our model hints at the latter, and suggests this scaling may be qualitatively different from scaling the width. On its own, it does not provide a complete explanation applicable in practice, but we hope it informs future work on the difference in the scaling behavior of vocabulary size and width.
>
> ## Why $\alpha < 1$?
> > Can you provide stronger justification or evidence from practical datasets (beyond Figure 3) that data following α < 1 is worth studying, and thus your results may be practically meaningful?
>
> We agree that the case $\alpha < 1$ is not the most relevant for practice.
> We describe it to illustrate that the case $\alpha = 1$ is a "worst-case" scenario. The decrease in performance of GD depends both on having heavy tails and a large imbalance in the frequencies. The case $\alpha > 1$ lacks the heavy tails, and the case $\alpha < 1$ lacks the imbalance. We will clarify this in the introduction.
>
> ## Hyperparameter implications?
> > Hyperparameter implications: [...] can you elaborate precisely on how practitioners might use these scaling laws to adjust specific hyperparameters
>
> Our scaling laws provide guidelines to set hyperparameters for the linear bigram model.
> GD requires the step-size to scales with the vocabulary size $d$ as $\eta \sim \log(d)$. SD requires the step-size to decrease with the step budget $T$ as well (Definition 4.4). While these scaling are likely not directly applicable to LLMs and would need to be combined with other approaches such as muP, those approaches do not yet consider the scaling of the step-size of the first layer as a function of the vocabulary size.
>
> ## Generalizability beyond the model
> > Generalizability: How do you anticipate your results might generalize beyond linear bigram models?
> > Which of your assumptions are most critical and least likely to hold in more complex models?
>
> The most crucial assumption is the linearity of the model. We do not expect our model to make quantitative testable predictions for LLM training, as many non-linear phenomenon cannot be captured by our model. We do however expect LLMs to exhibit a similar qualitative behavior in terms of training difficulty scaling with $d$.
>
> We invite you to read [the response to Reviewer pUyz](https://openreview.net/forum?noteId=uPDWwfJLys), [Reviewer xbaC](https://openreview.net/forum?noteId=pIZjJafrGp),
> and [Reviewer 9d42](https://openreview.net/forum?noteId=3xCuAZLexY) for additional experiments and discussions on the generalization to the cross-entropy loss, the use of Adam instead of SD, and the axis-aligned structure of the model in the first and last layers, respectively.
>
> ## Recommendations for readability
>
> We thank you for your suggestions on how to improve the readability of the paper.
> We will use the additional page for the camera ready to incorporate these suggestions
> (scaling law, data vs. eigenvalues, format of the data $X, Y$, and more explicit definitions).
> "or" was indeed meant as "also known as", and the instructions will be removed.
>
> ---
>
> We remain available for further clarifications or other questions you may have.
>
> **References**
> - Velikanov and Yarotsky, 2022: arxiv/2202.00992
> - Tao et al., 2024: arxiv/2407.13623

---

> > ### Comment · Reviewer_G8T2 · 2025-07-31
> > **Response to rebuttal**
> >
> > I have read the rebuttal and it all makes sense to me.  I maintain my positive opinion of this work.

---

### Official Review · Reviewer_9d42 · 2025-06-13

**Clarity:** 3
**Significance:** 3
**Originality:** 3
**Rating:** 4
**Confidence:** 4

**Summary:**

In this paper, the authors consider a linear bigram model, where the distribution of the current token and the conditional
distribution of the next token both follow a power law. They study the loss curve of gradient descent and sign descent
in this setting, and derive loss-iteration scaling laws that take the dimensional dependency into consideration and
are more accurate than the usual worst-case rate from the generic convex optimization analysis in the early phase of
the training. The scaling laws also show that when the data follow a power law distribution, sign descent has a
better dimensional dependency than vanilla gradient descent.

**Questions:**

* See the weakness section. For the sign descent analysis, what will happen if we apply a rotation to the input embedding?
  What if the embeddings are only near orthogonal (for example, random Gaussian vectors)? Do the analysis/result/intuition
  carry over to those cases?
* Could you provide some intuitive explanations on why sign descent has a better rate than gradient descent in this task?
* What will happen if the condtional/unconditional distributions have different exponents? I guess we will not be able
  to get clean formulae in this case. However, I wonder given a scaling law formula, whether it is always possible to find exponents
  $\alpha_k$ such that gradient descent or sign descent realizes this scaling law, or this model can realize only a
  certain class of scaling laws? I am asking this purely out of curiosity and do not expect a complete answer.

**Ethical Concerns:**

["NO or VERY MINOR ethics concerns only"]

**Final Justification:**

Though the paper considers relatively simple models, the analysis is neat and contains some interesting ideas (choosing a proper scaling between $t$ and $d$ to obtain a non-trivial asymptotic result).

My main concern was the axis-aligned assumption. In the rebuttal, the authors admitted that this is a strong assumption and cannot be removed, but they justify this assumption with existing empirical results showing this property can indeed be observed in real-world models, at least to a certain degree. I believe this is an acceptable justification.

Therefore, I recommend acceptance of the paper.

**Limitations:**

yes

**Quality:**

3

**Strengths And Weaknesses:**

**Strength**
* This is a neat paper in the sense that the task itself (bigram model) is simple, but the analysis is clean
  and complete. The overall presentation is also clear and easy to follow.
* The introduction of the rescaled time variable is new in the theoretical analysis of scaling laws (as far as I know).
  The authors let the time $t$ be a function of the dimension $d$, so that the loss $L_d(t)$ does not degenerate when
  $d \to \infty$. It is surprising that such a simple trick can lead to a wide variety of non-trivial results.
* The theoretical analysis is quite complete, as the authors derive the scaling laws for both gradient descent and sign
  descent in fo all different regimes of the power law exponent $\alpha$. Experiments/simulation are also provided to
  verify their theoretical claims.

**Weakness**
* The inputs are one-hot encoding vectors of the tokens, which are automatically axis-aligned. This greatly simplifies
  the analysis of sign descent. It is not clear if the sign descent analysis can be extended to other orthogonal (rotated version of the one-hot embeddings) or
  near orthogonal (random Gaussian vectors) embeddings. Though one may argue that algorithms such as Muon will remove the effect of a rotation, and it
  behaves similar to axis-aligned sign descent, this is not obvious and does not apply to the near orthogonal case.

**Typos and suggestions**
* Line 144: "$\pi_{\rho_j(k+1)|j} \ge \pi_{\rho_j(k)|j}$." I believe the inequality should be reversed.
* Line 225: "$\eta_* = \min_\eta...$" Should be $\mathrm{argmin}_\eta$
* Line 954: "At she partial sum ..."
* I found Proposition 2.2 confusing until I reached the proof Theorem 3.1. It might be good if the authors could comment
  near Proposition 2.2 that this proposition is intended to be used together with Eq (1) to get a clean formula for the
  loss.

---

> ### Author Rebuttal · Authors · 2025-07-30
>
> # General response
>
> We thank you for your efforts in this review and for the constructive feedback that will improve the paper.
>
> Multiple reviewers raised similar points regarding the simplicity of the model. We start by addressing those points, and then answer the questions specific to your review. We invite you to read the response to the other reviews for more information on the following points.
> - [Response to Reviewer pUyz](https://openreview.net/forum?noteId=uPDWwfJLys): **Cross-entropy loss vs. squared loss**
>   Capturing the dynamics of optimization on the cross-entropy loss is not possible, but experiments show that the performance gap between gradient descent (GD) and sign descent (SD) as $d$ increases is similar.
> - [Response to Reviewer xbaC](https://openreview.net/forum?noteId=pIZjJafrGp): **Averaging oscillations with Assumption 4.1 and Adam vs. sign descent**
>   SD is a simple proxy for Adam, but does capture the main benefit of a better scaling with $d$. Experimentally, Adam performs better than SD and also scales better than GD in $d$.
> - [Response to Reviewer 8S6d](https://openreview.net/forum?noteId=Ezuh3XcjFB): **Relation to prior scaling laws**
>   The main novelty of our approach is on the scaling of $t$ with $d$ to derive a scaling law for GD when the difficulty of the problem scales with $d$.
> - [Response to Reviewer 9d42](https://openreview.net/forum?noteId=3xCuAZLexY): **Axis-aligned model and sign descent**
>   The axis-aligned property of the model is a key assumption that allows not only for the analysis of SD but is also crucial to get an improvement over GD, and this property mimics the axis-alignment found in the first and last layers of LLMs.
> - [Response to Reviewer G8T2](https://openreview.net/forum?noteId=CfxXnAH6I3): **Connections to LLM training**
>   Discussion of the bigram frequencies for real tokenizers and possible insights on LLM training.
>
> We ran additional experiments to answer those questions, but as the NeurIPS guidelines do not allow sharing figures or updating the paper, we can only share a written description. We will use the additional page given for camera ready to include those experimental results and incorporate the clarifications and feedback provided by the reviewers.
>
> ## Simplicity of the model
>
> > Training language models uses softmax cross‑entropy, not squared error. [...]
> > _Reviewer pUyz_
> > Some of the assumptions are fairly unrealistic. Eg, the model is linear, uses square loss, assumes deterministic full-batch updates, and collapses Adam to SD [...]
> > _Reviewer xbaC_
> > the analysis is restricted to the simple linear bigram model with squared loss [which] may not directly generalize to complex transformer layers or cross-entropy objectives [...]
> > _Reviewer G8T2_
>
> Multiple reviewers have noted that the model we study is simple, and perhaps too simple to be of practical use for the training of LLMs.
> We agree with this assessment.
> We do not claim that our theory is directly applicable to LLMs.
> Understanding the impact of other hyperparameters is an interesting next step.
> The choice of loss function, mini-batch size, momentum, model (attention or normalization layers), or the study of generalization instead of training error are all likely to yield new insights.
> However, we believe the simplicity of the model is a strength of the paper.
>
> Despite its simplicity, the model is not trivial and already captures a complex behavior observed in practice, that SD outperforms GD for language data.
> Its simplicity allows us to establish clean theoretical results for the convergence of GD and SD without relying on random matrix theory, Fourier transforms, or other complex mathematical tools. The scaling laws provide an explanation for the improvement of SD over GD and ties it to the properties of the data through the heavy tails in the distribution of the eigenvalues. These results prove that the performance gap grows with the vocabulary size even on a linear regression model.
>
> We have ran additional experiments to show that the conclusion that SD scale better than GD extend to the cross-entropy loss and the use of Adam instead of SD. For space reasons, we cannot include these results in every response. We invite you to read [the response to Reviewer pUyz](https://openreview.net/forum?noteId=uPDWwfJLys) for more details on the cross-entropy loss, and [the response to Reviewer xbaC](https://openreview.net/forum?noteId=pIZjJafrGp) for more details on Adam.
>
> ---
>
> # Points raised in your review
>
> We now address the specific questions raised in your review.
>
> ## Axis-aligned model and why SD works
> > The inputs are one-hot encoding vectors of the tokens, which are automatically axis-aligned. This greatly simplifies the analysis of SD. [...] What will happen if we apply a rotation to the input embedding? What if the embeddings are only near orthogonal (for example, random Gaussian vectors)? Do the analysis/result/intuition carry over to those cases?
>
> > Could you provide some intuitive explanations on why SD has a better rate than GD in this task?
>
> We agree that the axis-aligned structure of the problem is a key part of the analysis, and not only a simplifying assumption. Adding a random rotation to the problem breaks the axis-aligned property, makes SD much worse than GD empirically, and makes the analysis much harder. We will revise the paper to emphasize the following.
>
> The benefits of SD come specifically from this axis-aligned structure. That SD has similar performance benefits as Adam in transformers has been shown experimentally by Kunstner et al. (2023, 2024), and the benefits of Adam have been shown experimentally to primarily come from how it handles in the first and last layer of the transformer by Zhang et al. (2024) and Zhao et al. (2024). Those layers are axis-aligned, and this structure is necessary to obtain good performance with SD, as the performance of Adam depends on this axis-alignment (Maes et al., 2024).
>
> Without additional assumptions on the axis-aligned structure, existing analyses predict that the performance of SD scales worse with dimension than plain GD (we compare our results with existing rates for SD. Our work shows the benefits of this axis-aligned structure on a problem that is close to the first layer of a transformer, and the benefits of SD in this setting.
>
> ## What if the exponents differ in $\pi_i$ and $\pi_{j\vert i}$?
> > What will happen if the condtional/unconditional distributions have different exponents?
>
> Given different exponents for the unconditional and conditional distributions, it should still be possible to obtain a scaling law for the loss. Although the number of cases with distinct functional forms might grow from the three currently covered ($\alpha < 1, \alpha=1, \alpha> 1$) to potentially more. The analysis of each case is likely not much more complex than the cases we cover, but presenting the results in a unified way may be difficult.
>
> > I wonder given a scaling law formula, whether it is always possible to find exponents $\alpha_k$ such that GD or SD realizes this scaling law, or this model can realize only a certain class of scaling laws?
>
> It is likely not possible to fit _any_ scaling law. Our model is flexible enough to cover multiple functional forms, including the three cases below for GD (taking $r = \frac{L_t - L_*}{L_0 - L_*}$)
> $$
>     \begin{aligned}
>     r \sim \frac{1-\alpha}{\alpha} \frac{\exp(\frac{2t}{d^{\alpha}})}{\frac{2t}{d^\alpha}+1}
>     \quad (\alpha < 1)
>     \quad &&\quad
>     r \sim 1 - \frac{\log(t)}{\log(d)}
>     \quad (\alpha = 1)
>     \quad &&\quad
>     r \sim \frac{C(\alpha) }{t^{1-\frac{1}{\alpha}}}
>     \quad (\alpha > 1)
>     \end{aligned}
> $$
> Adding a separate exponent for the conditional distribution would allow us to cover more cases,
> likely any power-law scaling of the form $L \sim t^{-c}$ form some $c$. But those are only three functional form, and it is likely that many other could be obtained by studying other models.
>
> ## Typos and suggestions
>
> Thank you for the recommendations, we will incorporate them.
>
>
> ---
>
> We remain available for further clarifications or other questions you may have.
>
> **References**
> - Kunstner et al., 2023: arxiv/2304.13960
> - Kunstner et al., 2024: arxiv/2402.19449
> - Maes et al., 2024: arxiv/2410.19964
> - Zhang et al., 2024: arxiv/2406.16793
> - Zhao et al., 2024: arxiv/2407.07972

---

> > ### Comment · Reviewer_9d42 · 2025-08-02
> >
> > I thank the authors for the clarifications. I'll maintain my positive score (4).

---

### Official Review · Reviewer_8S6d · 2025-07-02

**Clarity:** 3
**Significance:** 3
**Originality:** 3
**Rating:** 5
**Confidence:** 4

**Summary:**

This paper studies the optimization scaling law for GD and sign GD on a linear bigram model with square loss. They posit a power law distribution for the frequency of each word $\pi_k \sim k^{-\alpha}$. They provide scalings for the MSE loss as a function of $t$ for both GD and sign GD. One key takeaway for Zipfian data $\alpha = 1$ is that GD requires $d^{1-\epsilon}$ steps while sign GD requires $\sqrt d$ steps to converge where $d$ is the vocabulary size. The authors verify their approximations and scaling results with numerical simulations. Their work establishes the importance of optimizer in improving efficiency of training on large vocabularies with Zipfian structure.

**Questions:**

Questions

1. In the related works section, the authors claim that previous scaling law studies required $\alpha \geq 1$ in order to have well defined infinite width and depth limits. I think both the Paquette paper and the Bordelon scaling law paper could consider either $\alpha > 1$ or $\alpha \leq 1$ since their formulas depended on a general set of limiting eigenvalues. Could the authors explain what they meant in this section?
2. The observed scaling laws often do have the form of $\text{Loss} \sim t^{- \alpha}$. For GD this would require having $\alpha > 1$ to get this functional form and it could allow for a variety of observed scaling law relationships. From Theorem 4.5 a power law scaling for time $t$ is achieved for any $\alpha > \frac{1}{2}$ where it goes as $t^{-2}$. Empirically the LLM scaling laws have an exponent for time of $\approx 0.75$ (Kaplan). Do the authors think that this is compatible with their model?
3. Have the authors experimented with LLM training using varying tokenizers? The theory suggests that time to train should depend on $\sqrt{d}$. Validating this (or perhaps pointing to existing experiments on this) would be very impressive and strengthen the paper significantly.
4. Do the authors have a sense of what finite model size scaling laws look like in this model? If they added an untrainable projection layer $\hat{Y} = X A W$ where $A$ is $d \times k$ (could be random and frozen as in Bordelon or Paquette papers) and $W$ is now $k \times d$ can they describe the dependence of the final error on $k$ if $k < d$? Does it recover reasonable scaling laws (ie consistent with LLM papers?)

Nits

1. "would" on line 34 should probably be "could"
2. Should $\lambda_{ij}$ under line 129 be $\lambda_i$?

**Ethical Concerns:**

["NO or VERY MINOR ethics concerns only"]

**Limitations:**

The authors address limitations of the present study.

**Quality:**

3

**Strengths And Weaknesses:**

Strengths
1. This paper studies an important problem of analyzing neural scaling laws for sign-GD.
2. They provide a novel analysis of sign-GD on a bigram model and show that for Zipfian data, sign-GD outscores SGD significantly as a function of the token dimension ($\sqrt d$ instead of $d$ steps).
3. They also include several experiments of their bigram model showing the validity of their analytical results.

Weaknesses
1. Connections to known scaling law results for LLMs could be improved or clarified (see questions below).

---

> ### Author Rebuttal · Authors · 2025-07-30
>
> # General response
>
> We thank you for your efforts in this review and for the constructive feedback that will improve the paper.
>
> Multiple reviewers raised similar points regarding the simplicity of the model. We start by addressing those points, and then answer the questions specific to your review. We invite you to read the response to the other reviews for more information on the following points.
> - [Response to Reviewer pUyz](https://openreview.net/forum?noteId=uPDWwfJLys): **Cross-entropy loss vs. squared loss**
>   Capturing the dynamics of optimization on the cross-entropy loss is not possible, but experiments show that the performance gap between gradient descent (GD) and sign descent (SD) as $d$ increases is similar.
> - [Response to Reviewer xbaC](https://openreview.net/forum?noteId=pIZjJafrGp): **Averaging oscillations with Assumption 4.1 and Adam vs. sign descent**
>   SD is a simple proxy for Adam, but does capture the main benefit of a better scaling with $d$. Experimentally, Adam performs better than SD and also scales better than GD in $d$.
> - [Response to Reviewer 8S6d](https://openreview.net/forum?noteId=Ezuh3XcjFB): **Relation to prior scaling laws**
>   The main novelty of our approach is on the scaling of $t$ with $d$ to derive a scaling law for GD when the difficulty of the problem scales with $d$.
> - [Response to Reviewer 9d42](https://openreview.net/forum?noteId=3xCuAZLexY): **Axis-aligned model and sign descent**
>   The axis-aligned property of the model is a key assumption that allows not only for the analysis of SD but is also crucial to get an improvement over GD, and this property mimics the axis-alignment found in the first and last layers of LLMs.
> - [Response to Reviewer G8T2](https://openreview.net/forum?noteId=CfxXnAH6I3): **Connections to LLM training**
>   Discussion of the bigram frequencies for real tokenizers and possible insights on LLM training.
>
> We ran additional experiments to answer those questions, but as the NeurIPS guidelines do not allow sharing figures or updating the paper, we can only share a written description. We will use the additional page given for camera ready to include those experimental results and incorporate the clarifications and feedback provided by the reviewers.
>
> ## Simplicity of the model
>
> > Training language models uses softmax cross‑entropy, not squared error. [...]
> > _Reviewer pUyz_
> > Some of the assumptions are fairly unrealistic. Eg, the model is linear, uses square loss, assumes deterministic full-batch updates, and collapses Adam to SD [...]
> > _Reviewer xbaC_
> > the analysis is restricted to the simple linear bigram model with squared loss [which] may not directly generalize to complex transformer layers or cross-entropy objectives [...]
> > _Reviewer G8T2_
>
> Multiple reviewers have noted that the model we study is simple, and perhaps too simple to be of practical use for the training of LLMs.
> We agree with this assessment.
> We do not claim that our theory is directly applicable to LLMs.
> Understanding the impact of other hyperparameters is an interesting next step.
> The choice of loss function, mini-batch size, momentum, model (attention or normalization layers), or the study of generalization instead of training error are all likely to yield new insights.
> However, we believe the simplicity of the model is a strength of the paper.
>
> Despite its simplicity, the model is not trivial and already captures a complex behavior observed in practice, that SD outperforms GD for language data.
> Its simplicity allows us to establish clean theoretical results for the convergence of GD and SD without relying on random matrix theory, Fourier transforms, or other complex mathematical tools. The scaling laws provide an explanation for the improvement of SD over GD and ties it to the properties of the data through the heavy tails in the distribution of the eigenvalues. These results prove that the performance gap grows with the vocabulary size even on a linear regression model.
>
> We have ran additional experiments to show that the conclusion that SD scale better than GD extend to the cross-entropy loss and the use of Adam instead of SD. For space reasons, we cannot include these results in every response. We invite you to read [the response to Reviewer pUyz](https://openreview.net/forum?noteId=uPDWwfJLys) for more details on the cross-entropy loss, and [the response to Reviewer xbaC](https://openreview.net/forum?noteId=pIZjJafrGp) for more details on Adam.
>
> ---
>
> # Points raised in your review
>
> We now address the specific questions raised in your review.
>
> ## $\alpha < 1$ and "finite-dimensional" models
> > the authors claim that previous scaling law studies required $\alpha \geq 1$ in order to have well defined infinite width and depth limits. I think both the Paquette paper and the Bordelon scaling law paper could consider either $\alpha \geq 1$ or $\alpha \leq 1$ [...] Could the authors explain what they meant in this section?
>
> We agree that this statement requires clarification. We will revise it to convey the following.
>
> Existing scaling laws work in regimes where the difficulty of optimization does not scale with dimension. If $L \sim t^{-c}$ for some constant $c$ without requiring $t$ to grow with $d$, even the high-dimensional limit is effectively as hard as a finite dimensional problem. In our setting, this is the $\alpha \geq 1$ regime.
> The works of Paquette et al. (2024) on the random feature model and Bordelon et al. (2024) on the teacher-student study generalization and use different parameterizations, making a direct comparison difficult, but also fall in this "effectively finite-dimensional" regime.
> - In the work of Paquette et al. (2024), this can be seen from their phase diagrams in Figure 2.a and Figure 4.a. The area with very heavy tails in the Hessian eigenvalue and target does not admit a power-law decay, and likely does not decay without scaling $t$ with $d$.
> - Similarly, Bordelon et al. (2024) provide a scaling law of $L \sim t^{-(a-1)/b}$ (Eq. 4, in their notation) when the dataset and model size $\to\infty$. The case $a = 1$ corresponds to Zipf-law decay of the eigenvalues, and the loss is independent of $t$. For $a < 1$, the loss explodes instead. Those degenerate problems appear because they take the dimensionality of the model to infinity first and keep $t$ finite.
>
> In both cases, considering a joint limit where $t$ and $d$ scale together could capture interesting dynamics and show training can make progress, if only slowly.
>
> ## Comparison with empirical scaling laws
> > The observed scaling laws often do have the form of $L \sim t^{-\alpha}$. For GD this would require having $\alpha > 1$ to get this functional form and it could allow for a variety of observed scaling law relationships. [...] ]Empirically the LLM scaling laws have an exponent for time of $\approx 0.75$ (Kaplan). Do the authors think that this is compatible with their model?
>
> The comparison appears difficult, as the dimension taken to $\infty$ is not the same. Experimental works on scaling laws that achieve $L\sim t^{-3/4}$ increase parameter count by scaling the hidden dimension of the network. Given the same vocabulary size, it is likely that this makes the problem easier to fit rather than harder. This scaling is likely in the "effectively finite-dimensional" regime discussed above, and likely better modeled by existing analyses. The remaining differences could be explained by other effects, even without considering a more complex non-linear model; how to scale mini-batch size, momentum, and an analysis of the generalization error instead of training error are likely to have a significant impact on the exponent of the power in the power-law.
>
>
> ## Experimental evidence for $\sqrt{d}$ scaling
> > Have the authors experimented with LLM training using varying tokenizers? The theory suggests that time to train should depend on $\sqrt{d}$. Validating this (or perhaps pointing to existing experiments on this) would be very impressive and strengthen the paper significantly.
>
> The closest evidence for a scaling in vocabulary is the work of Tao et al. (2024), which shows experimentally that larger FLOPS budget benefit from larger vocabulary sizes. The analysis in FLOPS makes it unclear whether the need for more computation is due a higher cost per iteration or the need for more iterations however.
>
> We agree that an experimental result trying to establish non-linear model also exhibit a different scaling in vocabulary size with GD vs. SD would strengthen the paper. We will try to train small language models with varying vocabulary sizes, but do not have the computational resources to do so with models considered practical.
>
> ## Untrainable projection layer
> > Do the authors have a sense of what finite model size scaling laws look like in this model? If they added an untrainable projection layer $\hat Y = XAW$ where $A$ is $d \times k$ [random and frozen] and $W$ is now $k \times d$ can they describe the dependence of the final error on $k$ if $k < d$?
>
> The performance of sign descent will degrade considerably due to the removal of the axis-aligned structure of the problem and the scaling in $d$ will likely be much worse than for GD (see also the [response to Reviewer 9d42](https://openreview.net/forum?noteId=3xCuAZLexY) on the axis-aligned property of the model). If the dimension of the model $k$ scales with the vocabulary size $d$ and $k > d$ as in the work of Paquette et al. (2024), we suspect little change in the performance of GD on the expected error because of rotational invariance. The scaling with $k < d$ could be obtained by estimating the decay of the eigenvalues of the resulting Hessian random matrix, but we have not done the analysis.
>
> ---
>
> We remain available for further clarifications or other questions you may have.
>
> **References**
> - Bordelon et al., 2024: arxiv/2402.01092
> - Paquette et al., 2025: arxiv/2405.15074
> - Tao et al., 2024: arxiv/2407.13623

---

> > ### Comment · Reviewer_8S6d · 2025-08-04
> >
> > I thank the authors for their detailed rebuttal. I now have a better understanding of the relationship between their work and prior scaling law theories. I maintain my positive score and vote for acceptance.

---

### Official Review · Reviewer_xbaC · 2025-07-04

**Clarity:** 3
**Significance:** 2
**Originality:** 2
**Rating:** 4
**Confidence:** 3

**Summary:**

This paper investigates why gradient descent struggles to train language models with large vocabularies, focusing on how word frequency distributions affect optimization. The authors study a linear bigram model where word frequencies follow a power law (like Zipf's law in real text) and derive scaling laws showing how training time grows with vocabulary size.

Their main contributions are: (1) They prove that for Zipf-distributed text data (where word frequency is proportional to 1/rank), gradient descent requires nearly linear scaling with vocabulary size, explaining why GD performs poorly on language model embedding layers. In contrast, sign descent (a proxy for Adam) only needs iterations proportional to the square root of vocabulary size, providing up to 100x speedups for typical vocabularies. (2) They identify different scaling regimes based on how heavy-tailed the word distribution is: for very heavy-tailed data (including Zipf's law), the number of iterations must grow with dimension, while for lighter-tailed distributions, convergence is dimension-independent. (3) They develop a simplified model that captures the essential difficulty - frequent words dominate the loss but rare words in the long tail prevent fast convergence. Their theoretical predictions closely match experiments on real text data, providing a formal explanation for why Adam-like optimizers are crucial for training the first and last layers of transformers where vocabulary size matters most.

**Questions:**

- How well does sign descent capture the behavior of Adam in practice? An additional experiment which shows the convergence of Adam (or otherwise relaxing assumption 4.1) would strengthen the results.
- Similarly, how much do the other modeling assumptions change empirics? E.g. using mini-batches, alternative loss function, etc.?

**Ethical Concerns:**

["NO or VERY MINOR ethics concerns only"]

**Final Justification:**

I recommend acceptance. The paper is technically well executed, but there are questions around impact given the assumptions.

**Limitations:**

Yes

**Paper Formatting Concerns:**

Well formatted and organized. No concerns.

**Quality:**

3

**Strengths And Weaknesses:**

Strengths
- Strong theoretical analysis: rigorous asymptotic scaling laws for both gradient descent and sign descent on a well-motivated problem. Generally non-trivial analysis.
- Practical relevance: addresses a real phenomenon in deep learning - why Adam outperforms SGD on transformer embedding layers.
- Some of the simplifying assumptions are reasonably well motivated (eg Figure 3)
- Convincing empirical validation of the theoretical results
- Well organized and reasonable clarity
- Clear articulation of weaknesses

Weaknesses
- Some of the assumptions are fairly unrealistic. Eg, the model is linear, uses square loss, assumes deterministic full-batch updates, and collapses Adam to SD via an additional oscillation-averaging assumption (Assumption 4.1) whose fidelity is neither proved nor ablated.
- Empirical validation is still in a toy domain, and does not test how well the theory generalizes outside of the fairly restrictive modeling assumptions.
- The paper is fairly dense and would benefit from additional intuition.

[This is outside my area of expertise so I am unable to judge novelty and relation to prior work.]

---

> ### Author Rebuttal · Authors · 2025-07-30
>
> # General response
>
> We thank you for your efforts in this review and for the constructive feedback that will improve the paper.
>
> Multiple reviewers raised similar points regarding the simplicity of the model. We start by addressing those points, and then answer the questions specific to your review. We invite you to read the response to the other reviews for more information on the following points.
> - [Response to Reviewer pUyz](https://openreview.net/forum?noteId=uPDWwfJLys): **Cross-entropy loss vs. squared loss**
>   Capturing the dynamics of optimization on the cross-entropy loss is not possible, but experiments show that the performance gap between gradient descent (GD) and sign descent (SD) as $d$ increases is similar.
> - [Response to Reviewer xbaC](https://openreview.net/forum?noteId=pIZjJafrGp): **Averaging oscillations with Assumption 4.1 and Adam vs. sign descent**
>   SD is a simple proxy for Adam, but does capture the main benefit of a better scaling with $d$. Experimentally, Adam performs better than SD and also scales better than GD in $d$.
> - [Response to Reviewer 8S6d](https://openreview.net/forum?noteId=Ezuh3XcjFB): **Relation to prior scaling laws**
>   The main novelty of our approach is on the scaling of $t$ with $d$ to derive a scaling law for GD when the difficulty of the problem scales with $d$.
> - [Response to Reviewer 9d42](https://openreview.net/forum?noteId=3xCuAZLexY): **Axis-aligned model and sign descent**
>   The axis-aligned property of the model is a key assumption that allows not only for the analysis of SD but is also crucial to get an improvement over GD, and this property mimics the axis-alignment found in the first and last layers of LLMs.
> - [Response to Reviewer G8T2](https://openreview.net/forum?noteId=CfxXnAH6I3): **Connections to LLM training**
>   Discussion of the bigram frequencies for real tokenizers and possible insights on LLM training.
>
> We ran additional experiments to answer those questions, but as the NeurIPS guidelines do not allow sharing figures or updating the paper, we can only share a written description. We will use the additional page given for camera ready to include those experimental results and incorporate the clarifications and feedback provided by the reviewers.
>
> ## Simplicity of the model
>
> > Training language models uses softmax cross‑entropy, not squared error. [...]
> > _Reviewer pUyz_
> > Some of the assumptions are fairly unrealistic. Eg, the model is linear, uses square loss, assumes deterministic full-batch updates, and collapses Adam to SD [...]
> > _Reviewer xbaC_
> > the analysis is restricted to the simple linear bigram model with squared loss [which] may not directly generalize to complex transformer layers or cross-entropy objectives [...]
> > _Reviewer G8T2_
>
> Multiple reviewers have noted that the model we study is simple, and perhaps too simple to be of practical use for the training of LLMs.
> We agree with this assessment.
> We do not claim that our theory is directly applicable to LLMs.
> Understanding the impact of other hyperparameters is an interesting next step.
> The choice of loss function, mini-batch size, momentum, model (attention or normalization layers), or the study of generalization instead of training error are all likely to yield new insights.
> However, we believe the simplicity of the model is a strength of the paper.
>
> Despite its simplicity, the model is not trivial and already captures a complex behavior observed in practice, that SD outperforms GD for language data.
> Its simplicity allows us to establish clean theoretical results for the convergence of GD and SD without relying on random matrix theory, Fourier transforms, or other complex mathematical tools. The scaling laws provide an explanation for the improvement of SD over GD and ties it to the properties of the data through the heavy tails in the distribution of the eigenvalues. These results prove that the performance gap grows with the vocabulary size even on a linear regression model.
>
> We have ran additional experiments to show that the conclusion that SD scale better than GD extend to the cross-entropy loss and the use of Adam instead of SD. For space reasons, we cannot include these results in every response. We invite you to read [the response to Reviewer pUyz](https://openreview.net/forum?noteId=uPDWwfJLys) for more details on the cross-entropy loss, and [the response to Reviewer xbaC](https://openreview.net/forum?noteId=pIZjJafrGp) for more details on Adam.
>
> ---
>
> # Points raised in your review
>
> We now address the specific questions raised in your review.
>
> ## Averaging the oscillations in SD (Assumption 4.1) and comparison to Adam
> Multiple reviewers asked about the justification for the oscillation averaging in Assumption 4.1, and the connection between Adam and SD.
>
> > How well does SD capture the behavior of Adam in practice? An additional experiment which shows the convergence of Adam (or otherwise relaxing assumption 4.1) would strengthen the results.
>
> Assumption 4.1 is a simplification, but it is not overly strong. Once a component enters the oscillatory phase, we only miss-estimate its loss value by at most a factor of 2 every other step, so the real dynamics are within a constant factor of our model.
> This is a small price to pay for a great simplification, and not having to keep track of which component is on which side of the oscillation. An alternative would be to use a random initialization and use the fact that the oscillation should cancel out, but we do not think the benefits of this approach outweigh the added complexity.
>
> **On Adam**, we agree that the scaling law derived for SD is unlikely to hold for Adam. Our goal is to provide a simplified model that provably establishes the benefit of SD over GD, which has been idenfitied by prior work as a good proxy for the success of Adam (Kunstner et al., 2023, 2024). We run additional experiments to verify that our Adam also scales better with $d$ when $\alpha=1$. We fit the step-size by grid-search and fix other parameters to their default values ($\beta_1=0.9,\beta_2=0.999,\epsilon=10^{-8}$). The performance of Adam appears even better than that of SD, and does not appear to degrade as $d$ increases (or if it does, very slowly).
>
> ## Impact of changing the loss function
> > Similarly, how much do the other modeling assumptions change empirics? E.g. using mini-batches, alternative loss function, etc.?
>
> Capturing the dynamics of optimization on the cross-entropy loss is not possible, but experiments show that the performance gap between gradient descent (GD) and sign descent (SD) as $d$ increases is similar.
> We invite you to read the response to [Reviewer pUyz](https://openreview.net/forum?noteId=uPDWwfJLys) for a discussion of how the results change when using the cross-entropy loss instead of the squared loss.
>
> ---
>
> We remain available for further clarifications or other questions you may have.
>
> **References**
> - Kunstner et al., 2023: arxiv/2304.13960
> - Kunstner et al., 2024: arxiv/2402.19449

---

> > ### Comment · Reviewer_xbaC · 2025-08-03
> >
> > I appreciate your comments.
> >
> > I maintain my positive score and recommendation for acceptance. This is a technically solid work, though I still think the assumptions could be more thoroughly justified.

---

### Official Review · Reviewer_pUyz · 2025-07-05

**Clarity:** 2
**Significance:** 3
**Originality:** 3
**Rating:** 4
**Confidence:** 2

**Summary:**

This paper provides a theoretical analysis of why vanilla gradient descent (GD) slows down while Adam-like "sign descent" (SD) maintains performance when training embedding layers of language models with Zipfian token frequencies, using a tractable linear bigram next-token predictor with squared loss where both unigram and conditional distributions follow power laws π ∝ 1/k^α. The authors derive closed-form optimization dynamics for both GD and SD, establishing scaling laws that show the number of iterations t needed to reach fixed relative training error ε scales as d^α for GD (nearly linear in vocabulary size d for the realistic Zipf case α=1) but only as √d for SD when α≥½, yielding a substantial √d versus d^{1-ε} speedup advantage for SD over GD in the Zipfian regime. Through validation on synthetic power-law data and real OpenWebText bigram statistics, the analysis formally demonstrates that Zipfian spectra represent a "worst-case" scenario for gradient descent and provides theoretical justification for the empirical observation that Adam outperforms SGD on large-vocabulary language models.

**Questions:**

- Training language models uses softmax cross‑entropy, not squared error. Do the scaling exponents change when the loss is replaced by cross‑entropy on the same linear bigram model? A short derivation or experiment would clarify applicability.

**Ethical Concerns:**

["NO or VERY MINOR ethics concerns only"]

**Final Justification:**

I thank the authors for the response.
I have also read the other reviewer's comments. Overall i would keep my borderline accept rating.

**Limitations:**

Yes

**Quality:**

3

**Strengths And Weaknesses:**

Strengths
- Rigorous asymptotic derivations, with proofs sketched in the main text and deferred to appendices.
- Paper is well organised: assumptions ➔ theorems ➔ figures ➔ discussion
- Extends classical spectral‑decay analyses, which assumed α>1, to the heavy‑tailed case and to sign‑methods.

Weaknesses
- Core proofs rely on strong simplifications: squared loss, linear model, deterministic GD/SD;, momentum and cross‑entropy are ignored.

- The key SD analysis rests on Assumption 4.1 (replace oscillations by a constant η/2), whose correctness is argued only heuristically.

 - Because the model is linear, direct impact on full transformer training remains speculative.

---

> ### Author Rebuttal · Authors · 2025-07-30
>
> # General response
>
> We thank you for your efforts in this review and for the constructive feedback that will improve the paper.
>
> Multiple reviewers raised points regarding the simplicity of the model. We start by addressing those before answering your specific questions. We invite you to read the response to the other reviews for more information on the following points.
> - [Response to Reviewer pUyz](https://openreview.net/forum?noteId=uPDWwfJLys): **Cross-entropy loss vs. squared loss**
>   Capturing the dynamics of optimization on the cross-entropy loss is not possible, but experiments show that the performance gap between gradient descent (GD) and sign descent (SD) as $d$ increases is similar.
> - [Response to Reviewer xbaC](https://openreview.net/forum?noteId=pIZjJafrGp): **Averaging oscillations with Assumption 4.1 and Adam vs. sign descent**
>   SD is a simple proxy for Adam, but does capture the main benefit of a better scaling with $d$. Experimentally, Adam performs better than SD and also scales better than GD in $d$.
> - [Response to Reviewer 8S6d](https://openreview.net/forum?noteId=Ezuh3XcjFB): **Relation to prior scaling laws**
>   The main novelty of our approach is on the scaling of $t$ with $d$ to derive a scaling law for GD when the difficulty of the problem scales with $d$.
> - [Response to Reviewer 9d42](https://openreview.net/forum?noteId=3xCuAZLexY): **Axis-aligned model and sign descent**
>   The axis-aligned property of the model is a key assumption that allows not only for the analysis of SD but is also crucial to get an improvement over GD, and this property mimics the axis-alignment found in the first and last layers of LLMs.
> - [Response to Reviewer G8T2](https://openreview.net/forum?noteId=CfxXnAH6I3): **Connections to LLM training**
>   Discussion of the bigram frequencies for real tokenizers and possible insights on LLM training.
>
> We ran additional experiments to answer those questions, but as the NeurIPS guidelines do not allow sharing figures or updating the paper, we can only share a written description. We will use the additional page given for camera ready to include those experimental results and incorporate the clarifications and feedback provided by the reviewers.
>
> ## Simplicity of the model
>
> > Training language models uses softmax cross‑entropy, not squared error. [...]
> > _Reviewer pUyz_
> > Some of the assumptions are fairly unrealistic. Eg, the model is linear, uses square loss, assumes deterministic full-batch updates, and collapses Adam to SD [...]
> > _Reviewer xbaC_
> > the analysis is restricted to the simple linear bigram model with squared loss [which] may not directly generalize to complex transformer layers or cross-entropy objectives [...]
> > _Reviewer G8T2_
>
> Multiple reviewers have noted that the model we study is simple, and perhaps too simple to be of practical use for the training of LLMs.
> We agree with this assessment.
> We do not claim that our theory is directly applicable to LLMs.
> Understanding the impact of other hyperparameters is an interesting next step.
> The choice of loss function, mini-batch size, momentum, model (attention or normalization layers), or the study of generalization instead of training error are all likely to yield new insights.
> However, we believe the simplicity of the model is a strength of the paper.
>
>
> Despite its simplicity, the model is not trivial and already captures a complex behavior observed in practice, that SD outperforms GD for language data.
> Its simplicity allows us to establish clean theoretical results for the convergence of GD and SD without relying on random matrix theory, Fourier transforms, or other complex mathematical tools. The scaling laws provide an explanation for the improvement of SD over GD and ties it to the properties of the data through the heavy tails in the distribution of the eigenvalues. These results prove that the performance gap grows with the vocabulary size even on a linear regression model.
>
> We have ran additional experiments to show that the conclusion that SD scale better than GD extend to the cross-entropy loss and the use of Adam instead of SD. For space reasons, we cannot include these results in every response. We invite you to read [the response to Reviewer pUyz](https://openreview.net/forum?noteId=uPDWwfJLys) for more details on the cross-entropy loss, and [the response to Reviewer xbaC](https://openreview.net/forum?noteId=pIZjJafrGp) for more details on Adam.
>
> ---
>
> # Points raised in your review
>
> We now address the specific questions raised in your review.
>
> ## Quadratic vs. cross-entropy loss
>
> > Do the scaling exponents change when the loss is replaced by cross‑entropy on the same linear bigram model? A short derivation or experiment would clarify applicability.
>
> Theoretically, we cannot describe the dynamics of optimizers on the cross-entropy loss.
> Our analysis is restricted to the quadratic loss, and
> the closest existing analysis on similar models
> are restricted to continuous time (Cabannes et al., 2024, Kunstner et al., 2024)
> as the non-linearity of the gradient update make a discrete-step analysis intractable.
>
> However, we were able to confirm experimentally that
> the separation between GD and SD as the vocabulary size $d$ increases
> still holds for the cross-entropy loss for $\alpha = 1$.
> We pick the best step-size for a given time horizon by grid-search,
> and try to find the scalings by plotting.
> - The best step-size appears to scale as $\eta \sim 1/\log(d)^2$ for GD (instead of $\eta \sim 1/\log(d)$ for the quadratic loss)
> - The normalized optimality gap still appears to follow $r \sim 1 - \frac{\log(t)}{\log(d)}$ but converges more slowly to this limit as $d$ grows (or, possibly, to another limit).
> - The normalized optimality gap for SD appears independent of $d$ for the range of $d$ we were able to test run.
>
> These results suggest that the performance gap between GD and SD still grows with $d$ for the cross-entropy loss, though the exact form of the scaling law may be different.
>
> ## Averaging the oscillations in SD (Assumption 4.1)
> > The key SD analysis rests on Assumption 4.1 (replace oscillations by a constant η/2), whose correctness is argued only heuristically.
>
> SD is a simple proxy for Adam, but does capture the main benefit of a better scaling with $d$. Experimentally, Adam performs better than SD and also scales better than GD in $d$. We invite you to read the response to [Reviewer xbaC](https://openreview.net/forum?noteId=pIZjJafrGp) for a discussion of Assumption 4.1.
>
> ---
>
> We remain available for further clarifications or other questions you may have.
>
> **References**
> - Cabannes et al., 2024: arxiv/2402.18724
> - Kunstner et al., 2024: arxiv/2402.19449

---

> > ### Comment · Reviewer_pUyz · 2025-08-03
> > **Thanks for your response**
> >
> > Thank you for your response. I will keep my positive score.

---

### Decision · Program_Chairs · 2025-09-17

**Decision:**

Accept (poster)

**Comment:**

The paper explores scaling laws on a simple linear model with power-law covariance, $\lambda_j = 1/j^{\alpha}$. In particular, this work focuses on the setting where the exponent is $\alpha < 1$. The authors derive the scaling laws for both gradient descent and sign descent. The paper is well-written with rigorous theoretical analysis. All reviewers were fairly positive about the work and thought it was interesting.  While the work is on a simplistic setting, it provides insights and it is a stepping stone to understanding scaling laws of more complex objectives. Overall this is a solid submission.